# Nuclear lamina strain states revealed by intermolecular force biosensor

Brooke E. Danielsson [1,6], Bobin George Abraham [2,6], Elina Mäntylä [2], Jolene I. Cabe[1], Carl R. Mayer[1], Anna Rekonen[2], Frans Ek [2], Daniel E. Conway [3,4] ✉ & Teemu O. Ihalainen [2,5] ✉

Nuclear lamins have been considered an important structural element of the nucleus. The nuclear lamina is thought both to shield DNA from excessive mechanical forces and to transmit mechanical forces onto the DNA. However, to date there is not yet a technical approach to directly measure mechanical forces on nuclear lamins at the protein level. To overcome this limitation, we developed a nanobody-based intermolecular tension FRET biosensor capable of measuring the mechanical strain of lamin filaments. Using this sensor, we were able to show that the nuclear lamina is subjected to significant force. These forces are dependent on nuclear volume, actomyosin contractility, functional LINC complex, chromatin condensation state, cell cycle, and EMT. Interestingly, large forces were also present on nucleoplasmic lamins, indicating that these lamins may also have an important mechanical role in the nucleus. Overall, we demonstrate that the nanobody-based approach allows construction of biosensors for complex protein structures for mechanobiology studies.

Mechanical forces are important co-regulators of many physiological processes[1]. In addition to mechanotransduction at the surface of the cell, the cytoskeleton also allows transmission of forces throughout the cell, including onto and within the nucleus[2]. Thus, the nucleus has emerged as a putative mechanosensitive structure. At the nuclear envelope (NE), Linker of Nucleoskeleton and Cytoskeleton (LINC) complex, consisting of nesprin and SUN proteins, form the principal structure that connects the nucleus to the cytoskeleton[3]. These connections enable a mechanotransmission pathway, where mechanical stress can be transduced bidirectionally between the cell surface and the nucleus via the cytoskeleton[4]. Inside the nucleus LINC complex connects to nuclear lamina, a protein rich meshwork lining the inner nuclear membrane. The lamina is ~15-nm-thick protein meshwork, formed mainly from flexible ~400-nm-long A-type and B-type lamin filaments[5,6]. Large parts of the chromatin are tethered to the nuclear

lamina and this interaction has been shown to regulate gene expression[5]. Especially A-type lamin proteins, lamin A/C, are also located throughout the nucleoplasm. These nucleoplasmic lamins bind to chromatin and have been indicated to regulate chromatin accessibility and spatial chromatin organization[7]. Furthermore, mechanical stress has been shown to be transmitted deep into the nucleus and affect nuclear substructures, and even local chromatin organization and transcription in lamin A/C dependent manner[2,8]. Thus, nuclear lamina forms a physical interface between chromatin and cytoplasm, and this interface is exposed to different mechanical cues.

In vitro experiments of purified nuclear lamins have shown that these proteins are able to withstand large mechanical forces, exhibit deformation under mechanical loading, and show strain-stiffening behavior[9]. Inside the cells the amount of A-type lamins depends on cell

[1]Department of Biomedical Engineering, Virginia Commonwealth University, Richmond, Virginia, USA. [2]BioMediTech, Faculty of Medicine and Health Technology, Tampere University, Tampere, Finland. [3]Department of Biomedical Engineering, The Ohio State University, Columbus, Ohio, USA. [4]The Ohio State University and Arthur G. James Comprehensive Cancer Center, The Ohio State University, Columbus, Ohio, USA. [5]Tampere Institute for Advanced Study, Tampere University, Tampere, Finland. [6]These authors contributed equally: Brooke E. Danielsson, Bobin George Abraham. ✉e-mail: conway.362@osu.edu; teemu.ihalainen@tuni.fi

substrate rigidity and nuclear lamina organization is affected by cellular contractility[10,11]. Currently, there are no methods to directly detect changes in lamina strain state and this precludes studies on the effect of lamin strain-state in mechanotransduction between cytoskeleton and nucleus and its downstream implications. To study the mechanical loading of lamin filaments in vivo with subcellular resolution, we sought to develop a biosensor for lamin A/C. Prior force biosensor design strategies consisted of chimeric proteins in which a FRET pair separated by a force-sensitive peptide was inserted in the middle of the protein[12]. These intramolecular force sensors have been successfully developed for proteins within focal adhesions[13–15], cell–cell adhesions[16–20], and the nuclear LINC complex[21–23]. However, concerns remain regarding how internal insertion of a large FRET-force module may affect the biological functions of the protein. This may be especially important in the context of filamentous proteins, such as the nuclear lamins, where an altered protein size and structure may impair the assembly of filamentous structures.

Here we introduce a nanobody-based intermolecular strain sensor concept for cellular mechanobiology studies. In this concept, instead of inserting a FRET-force module into the protein directly (intramolecular), we used nanobodies which bind to two proteins of interest (intermolecular). Our intermolecular strain sensor measures mechanical deformations between two proteins, in contrast to intramolecular force sensors which measure mechanical tension across a single protein. Our Lamin A/C strain sensor (Lamin-SS) consists of an existing FRET module (TsMod)[13] is flanked on each side with nanobodies targeted against Lamin A/C[24]. This indirect sensor design enables indirect tagging of endogenous proteins of interest without significantly impairing assembly, expression levels or cellular localization of these proteins. With this notion, we designed Lamin A/C strain sensor (Lamin-SS) using nanobodies targeted against Lamin A/C. Expressing this sensor in Madin-Darby canine kidney (MDCK) cells we demonstrate that this sensor exhibits an inverse FRET-force relationship. To validate the sensor, we used osmotic shock -induced nuclear shrinking and actomyosin contractility inhibitors. Using the sensor we are able to show that changes in chromatin condensation have a significant impact on lamin A/C force. Furthermore, our findings also show that the sensor experiences similar levels of tension in the nucleoplasm as compared to the nuclear lamina. The ability of nucleoplasmic lamin A/C to impart tensile forces was further supported by a second nanobody-based strain sensor which showed tensile forces between histone H2A-H2B and lamin A/C (Lamin-histone-SS) also demonstrating the versatility of the nanobody-based strain sensors. This technical advancement provides significant insights into nuclear mechanics, by providing the first direct measurements of nuclear lamina forces at the protein level. Additionally, the sensor design demonstrates the potential for nanobody-based biosensors to be further utilized to measure mechanical forces between proteins.

## Results

### Nanobody-based lamin A/C strain sensor measures mechanical force exerted by lamins

We developed a lamin A/C strain sensor (Lamin-SS) that consists of an existing FRET-force module, known as TSmod[13] with N- and C-terminal lamin A/C nanobodies[24] (Fig. 1a and Supplementary Fig. 1). TSmod is a well-established FRET-force module which has an inverse FRET-force relationship. In the sensor the force module was flanked by a flexible linker and nanobodies, which bind to their target epitope (Supplementary Fig. 1a). Nanobodies recognize a single binding site in the target protein, thus behaving like monoclonal antibodies. We mapped the lamin nanobody epitope by using competitive antibody binding assay and FRET experiments with truncated lamin A/C constructs. The binding of an antibody recognizing an epitope in the end segment of the coiled-coil domain of lamin A/C (rod-domain antibody EP4520)

was reduced in the presence of increasing concentration of Lamin-SS sensor (Supplementary Fig. 2a–c). FRET experiments with lamin A/C constructs N-terminally tagged with mScarlet and EGFP-conjugated lamin nanobodies showed higher FRET with truncates containing only the end part of the coiled-coil domain (Supplementary Fig. 2d, e). This indicates that nanobody was able to bind this region and this facilitated higher FRET between the mScarlet and EGFP. Together the data shows that the lamin nanobody binds to an epitope, which resides in the end of the coiled-coil domain.

Nanobody interaction with the target proteins has been reported to be strong, and the needed unbinding force for e.g., EGFP nanobody-epitope pair is tens of piconewtons[25]. These forces are considerably larger than the force sensitivity of the TSmod force module (most sensitive between 1 and 6 pN)[13], thus the sensor can be used to quantify the intermolecular force transduction. The molecular weight of the whole sensor is ~95.5 kDa (Supplementary Fig. 1a) and according to our model the physical length of the sensor is between 21 and 41 nm, depending on the sensor strain state (Supplementary Fig. 1b). The sensor length also defines the possible nanobody epitope-to-epitope distance to achieve simultaneous binding from both ends of the sensor. Based on the current understanding on lamin filament structure, the distance between the identical epitopes is ~20 nm in the relaxed lamin filament[6] (Supplementary Fig. 3a). Recently it has been speculated that the filament length can increase substantially under force, due to the tension induced sliding and unfolding of the lamin proteins within the filament[9]. Thus, the stretching of the filament can directly create strain to the Lamin-SS sensor. Furthermore, the lamins form a dense protein meshwork[6] and the nanobody epitopes can reside also in adjacent lamin A/C filaments (Supplementary Fig. 3b). The possible movement of the filaments can also create strain to the Lamin-SS sensor. Thus, the sensor can sense strain in the single lamin A/C filament or in the whole lamina network. In addition to Lamin-SS, a force-insensitive truncated control sensor (Lamin-TM), containing only an N-terminal lamin A/C nanobody, was also developed (Fig. 1a). Next, we generated MDCK cell lines stably expressing Lamin-SS or Lamin-TM. The fluorescence of both sensors was strongly correlated to lamin A/C immunostaining (Fig. 1b), indicating that the sensors have strong localization to the nuclear lamina. Analysis of apicobasal intensity ratio of Lamin-SS showed no difference in the binding of the sensor to apical or basal sides to the nuclear envelope, suggesting that the binding is not force sensitive (Supplementary Fig. 3c). Finally, western blotting against lamin A/C showed that the sensor expression did not influence the lamin A/C expression levels (Supplementary Fig. 1c).

First we measured the FRET of each sensor using spectral-based FRET measurements. Lamin-SS exhibited reduced FRET as compared to Lamin-TM (Fig. 1c) indicating an increased distance between the FRET pair (assumed to be due to tension across the elastic peptide in TSmod). Lamin-SS had a median sFRET efficiency of 18% (mean $0.18 \pm 0.01$ SEM) compared to Lamin-TM with 42% (mean $0.40 \pm 0.02$ SEM). As an additional control, we further confirmed these FRET changes using fluorescence-lifetime imaging microscopy (FLIM), which also showed reduced FRET (measured as increased lifetime) for Lamin-SS, as compared to Lamin-TM (Fig. 1d). To confirm that the FRET changes are dependent on A-type lamins, we constructed a MDCK cell line with CRISPR-Cas9 knockout (KO) of LMNA gene coding for A-type lamins. We characterized the LMNA KO in the MDCK cells by performing western blotting, immunofluorescence labeling against the N-terminus of lamin A/C and sequencing of the affected LMNA gene. Immunofluorescence experiments and western blotting indicated undetectable levels of lamin A/C in the LMNA KO cells (Supplementary Fig. 4a, c). Sequencing showed that there is a single base deletion 3 bp upstream of the protospacer adjacent motif site resulting in a frameshift in LMNA gene (Supplementary Fig. 4b). The frameshift mutation leads to scrambled protein amino acid sequence and emergence of an early stop codon, destroying the functionality of the translated

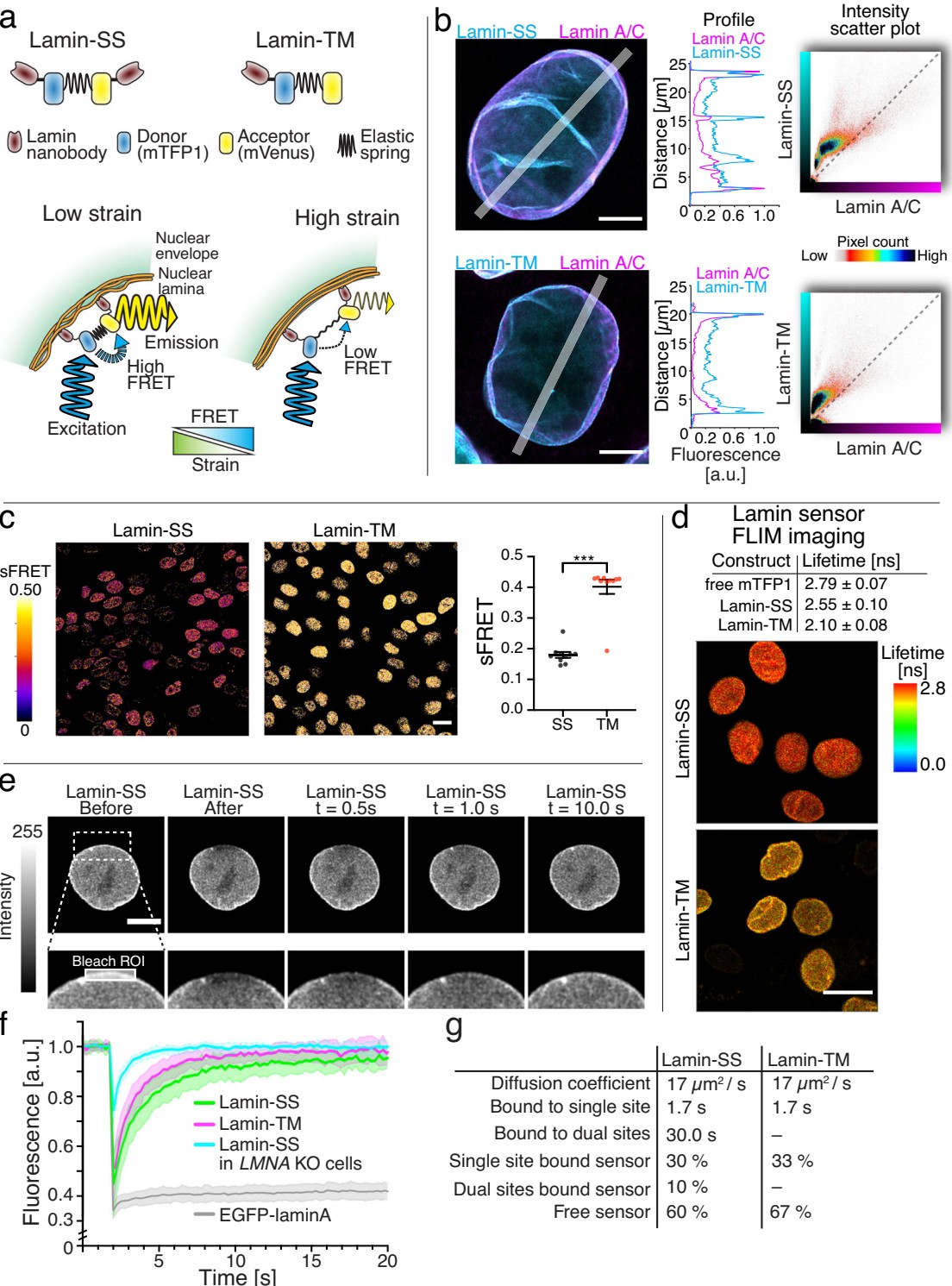

**Fig. 1 | Development and characterization of the FRET based lamin A/C strain sensor. a** Schematic representation of the FRET -based lamin A/C strain sensor (Lamin-SS), truncated control sensor (Lamin-TM) and the working mechanism of the strain sensing. **b** Representative confocal Airyscan xy-sections of immunolabeled lamin A/C (magenta) together with Lamin-SS or Lamin-TM (cyan) and corresponding fluorescence line-profiles ($n = 4$ and $n = 8$ cells, respectively, two biological replicates). Scatter-plot showing correlation of fluorescence intensities between the Lamin-SS or Lamin-TM and lamin A/C. Scale bars, 5 μm. **c** sFRET efficiency images and quantified sFRET (mean ± SEM) of Lamin-SS and Lamin-TM sensors ($n = 10$ fields, two biological replicates). Scale bar 20 μm. Unpaired two-tailed Student's $t$ tests (***$p < 0.0001$, $t = 8.8$, df = 18, $p = 0.0148$). **d** Donor fluorescence lifetimes of free donor (mTFP1), Lamin-SS and Lamin-TM along with FLIM images of Lamin-SS and Lamin-TM expressing cells ($n = 36$, $n = 43$ and $n = 42$ cells, two biological replicates). Scale bar 20 μm. **e** Example of a FRAP experiment with Lamin-SS expressing cell. Bleached region of interest (ROI) is marked in the blow-up image. Scale bar 5 μm. **f** Quantified and normalized fluorescence recoveries (mean ± STDEV) of Lamin-SS ($n = 18$), Lamin-TM ($n = 18$), and EGFP-lamin A ($n = 14$) in WT MDCK cells and Lamin-SS recovery ($n = 13$) in LMNA KO cells showing differences in the recovery dynamics (all the FRAP data from 1 to 3 biological replicates). **g** Lamin-SS and Lamin-TM binding times and corresponding fractions based on the simulated recoveries.

product of the LMNA gene. Loss of localization at the nuclear rim and higher FRET for Lamin-SS, quantified by using ratiometric FRET imaging (riFRET), were observed in lamin A/C knockout cells (Supplementary Fig. 4d-e), showing that the reduced FRET observed with Lamin-SS is dependent on A-type lamins. Median Lamin-SS riFRET was 16% (mean $0.157 \pm 0.007$ SEM) in WT and 19% (mean $0.197 \pm 0.005$ SEM) in LMNA KO cells. Additionally, fluorescence recovery after photobleaching (FRAP) experiments (Fig. 1e) of the sensor behavior showed that both sensors, Lamin-SS and Lamin-TM, bind to nuclear lamina. Lamin-TM showed fast recovery kinetics indicating rapid binding-unbinding dynamics of the truncated sensor to nuclear lamina (Fig. 1e). Lamin-SS recovery was considerably slower than that of Lamin-TM in wild-type MDCK cells, but much faster in LMNA KO cells (Fig. 1e, f). As a control, we also performed FRAP experiments to EGFP-lamin A expressing MDCK cells. Lamin A has been shown to bind strongly to the nuclear lamina[26]. In line with this, the FRAP experiments of EGFP-lamin A showed minimal recovery during the experiment time scale (Fig. 1f). Next, we quantified the sensor behavior by simulating the sensor binding to the nuclear lamina during the FRAP experiments (Supplementary Figs. 5 and S6). Since EGFP-lamin A FRAP experiments indicated that the Lamin A/C recovery was slow, we assumed in the simulations that the sensor binds immobile A-type lamins. Lamin-SS recovery in LMNA KO cells could be simulated with a single population of molecules diffusing freely with a diffusion coefficient of $17\,\mu m^2/s$ (Fig. 1g and Supplementary Fig. 5). In the subsequent simulations we used this diffusion coefficient for all the sensors and adjusted only binding parameters of the sensor. The recovery of the control sensor Lamin-TM in wild type MDCK could be replicated with two populations, one freely diffusing and the other showing binding with a short binding time (Fig. 1g and Supplementary Fig. 5). Finally, the dual nanobody sensor Lamin-SS showed additional bound fraction with substantially longer binding time to nuclear lamina (Fig. 1f, g and Supplementary Figs. 5 and 6).

## Lamin A/C forces are affected by changes in nuclear size and shape

To further validate FRET-force responsiveness of Lamin-SS, we measured the FRET ratio during osmotic shock-induced nuclear shrinkage. In agreement with prior work[27], hyperosmotic shock reduced nuclear volume (Fig. 2a, b), suggesting that nuclear lamina is mechanically more relaxed after the shock. Through paired cell analysis before and after sucrose-induced nuclear volume changes, we observed an increase in Lamin-SS FRET ratio, indicating less tensed sensor (Fig. 2c, d). Quantified Lamin-SS median riFRET efficiency was 6.2% (mean $0.064 \pm 0.001$ SEM) in control conditions (MEM) and 11.2% (mean $0.114 \pm 0.001$ SEM) in hyperosmotic conditions (MEM + 250 mM sucrose, 15 min).

Next, we wanted to increase the nuclear lamina strain and investigate its effect on Lamin-SS FRET. This was achieved by using polyacrylamide (PAA) hydrogel cushion to squeeze the cells and nuclei[11]. The cells were first briefly treated with actin fiber depolymerizing agent (cytochalasin D) to mechanically relax the cells and hinder the possible actin cytoskeleton reorganization following the compressive stress (Fig. 2e). This allowed us to study the direct effect of nuclear deformation on the Lamin-SS FRET. The drug treatment alone reduced lamin A/C forces, which was detected as increased FRET of the Lamin-SS (Supplementary Fig. 7). The median riFRET increased from 4.6% (mean $0.046 \pm 0.003$ SEM) before to 5.3% (mean $0.053 \pm 0.003$ SEM) after the cytochalasin D treatment, indicating direct role of actin cytoskeleton in nuclear lamina strain state. In the gel cushion experiments, nuclei were considerably stretched when the cells were under compressive stress for 15 min (Fig. 2e). The actin depolymerization prior the gel led to median riFRET ratio of 5.0% (mean $0.050 \pm 0.02$ SEM) and this was reduced to 4.3% (mean $0.043 \pm 0.001$ SEM) when the nuclei were strained by the gel (Fig. 2f, g).

## Lamin A/C forces are dynamic, responding to changes actomyosin contractility, and nuclear-cytoskeletal connectivity

To investigate the role of intracellular tension on forces in lamin A/C filaments, we sought to determine if the actomyosin contractility contributes to Lamin-SS FRET. When using Lamin-SS we observed that reduced actomyosin contractility via ROCK-pathway inhibitor (Y-27632) decreased lamin A/C forces (Figs. 3a and 2b). Median sFRET efficiency of Lamin-SS was 15% without treatment (mean $0.14 \pm 0.01$ SEM) and 22% with Y-27632 (mean $0.22 \pm 0.01$ SEM). Median sFRET efficiencies were 43% (mean $0.43 \pm 0.002$ SEM) and 42% (mean $0.39 \pm 0.02$ SEM) with Lamin-TM, respectively. In order to observe the overall effect of actomyosin contractility inhibition to the nucleus shape, we also quantified the nuclear circularity and cross-section area from the center plane of the nucleus. The data indicated that the inhibition did not affect the circularity, but with Lamin-TS expressing cells the cross-section area was reduced, suggesting mechanical relaxation of the nucleus (Supplementary Fig. 8). Furthermore, the sensor was successfully used to temporally analyze lamin A/C force changes during actomyosin inhibition. The timelapse imaging data indicated that the FRET increased rapidly after the addition of the drug and 20 minutes of treatment showed significantly increased relative riFRET (Fig. 3c, d). Thus, actomyosin contractility contributes to mechanical forces on lamin A/C filaments.

Because cytoskeletal tension is transduced to nuclear lamina in part by the nuclear LINC complex[28], we sought to understand the role of this structure for nuclear lamina forces. Disruption of the LINC complex using a dominant negative nesprin construct (DN-KASH)[28] modestly reduced lamin A/C forces (Fig. 3e, f), further indicating the role of the cytoplasmic cytoskeleton for forces applied to the lamin A/C network. Median sFRET efficiency of Lamin-SS was 22% without DN-KASH expression (mean $0.16 \pm 0.01$ SEM) and 28% (mean $0.19 \pm 0.01$ SEM) with DN-KASH expression. The efficiencies were 54% (mean $0.42 \pm 0.001$ SEM) and 53% (mean $0.40 \pm 0.01$ SEM) with Lamin-TM, respectively. The expression of DN-KASH slightly increased the nucleus circularity in Lamin-SS expressing cells, without influencing the cross-sectional area, suggesting that the LINC complex disruption did not affect the overall nuclear strain state in large extent (Supplementary Fig. 8). Taken together these data also demonstrate that lamin A/C forces are dynamic and they are affected by the cytoskeletal forces and intact LINC complexes. The data also demonstrates the dynamic responsiveness of the Lamin-SS sensor.

## Lamin A/C forces respond to cell cycle, EMT, and chromatin condensation

The detected FRET values of Lamin-SS differed substantially between the cells (Figs. 1–3). Due to this heterogeneity, we hypothesized that the cell-to-cell variations in cellular activities and physiological processes could be affecting lamin A/C forces. To investigate the physiological factors influencing lamin A/C forces we next manipulated the processes known to affect cell and nuclear mechanics by induction of cell cycle arrest, epithelial to mesenchymal transition (EMT), and reorganization of chromatin[29–31]. When cells were first arrested to early S-phase by treatment with DNA polymerase α inhibitor (aphidicolin), we detected increased Lamin-SS FRET (Fig. 4a), suggesting that lamin A/C forces are reduced by the cell cycle. Median FRET efficiency of Lamin-SS was 16% without Aphidicolin (mean $0.162 \pm 0.004$ SEM) and 21% (mean $0.208 \pm 0.004$ SEM) after Aphidicolin treatment. The efficiencies were 41% (mean $0.39 \pm 0.02$ SEM) and 42% (mean $0.39 \pm 0.02$ SEM) with Lamin-TM, respectively. However, the cell-to-cell variation in force persisted indicating that additional factors beyond cell cycle are creating this heterogeneity in forces.

One of the hallmarks of EMT is increased contractility of the cells, including actin stress fiber formation[30]. However, spatial analysis of forces has shown that this increased contractility does not increase tension in all regions of the cell. For example, we have shown that EMT

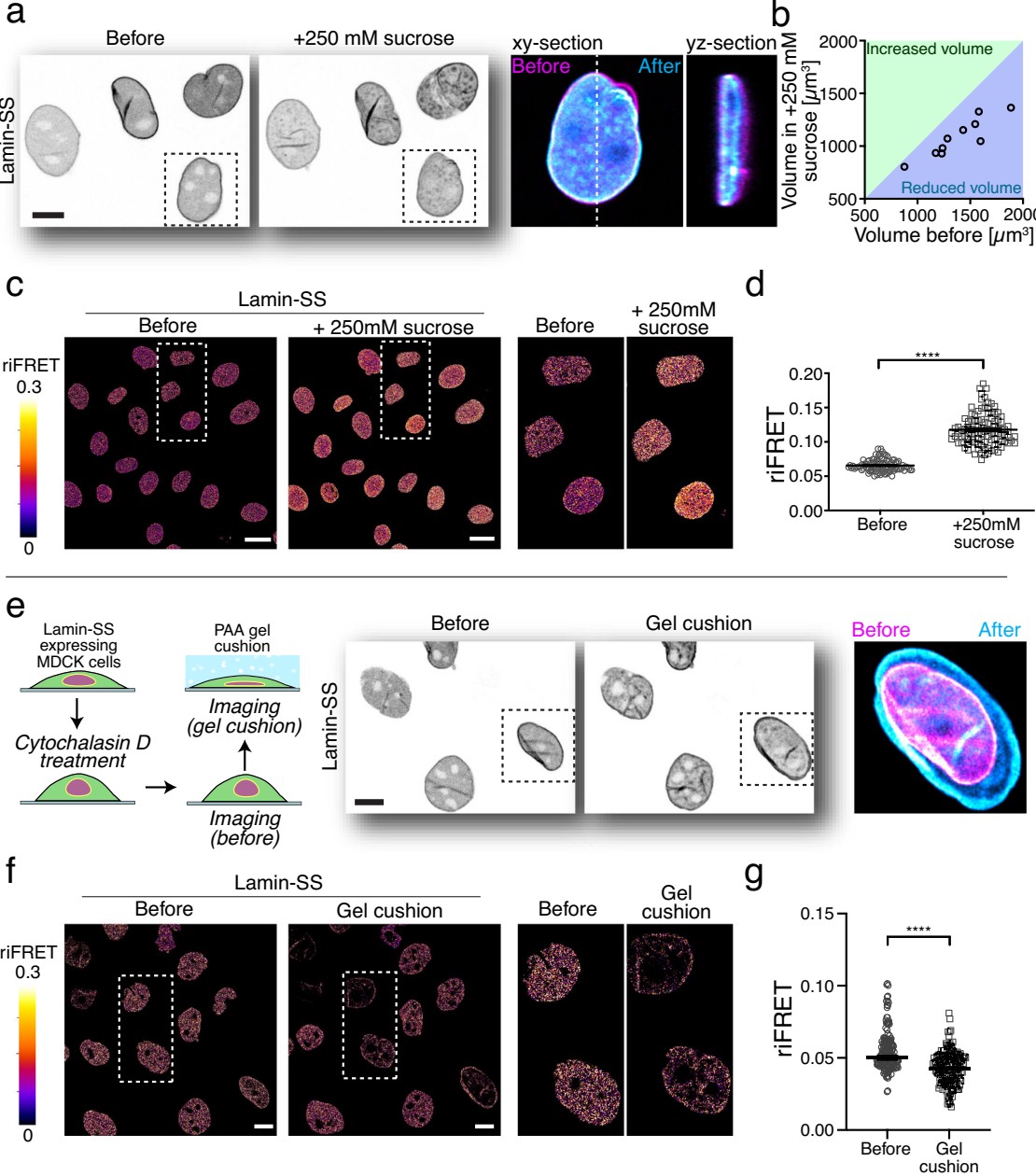

**Fig. 2 | The effect of nuclear deformation on lamin A/C strain state.**
**a** Representative image of Lamin-SS expressing cells (left) subjected to hyper-osmotic conditions by adding medium containing 250 mM sucrose for 15 min before the imaging (middle). Single cell blow-up indicated the change in nuclear morphology before (magenta) and after (cyan) the osmotic shock (right). Scale bar 10 μm. **b** Scatterplot of quantified nuclei volumes indicating clear reduction of the nuclear volume. **c** riFRET efficiency images of osmotically stressed Lamin-SS expressing cells and blow-up images. Scale bars, 20 μm. **d** Quantified Lamin-SS riFRET (mean ± SEM; $n$ = 304 cells, three biological replicates). Two-tailed Wilcoxon matched-pairs test (****$p$ < 0.0001). Scale bar 20 μm. **e** Hydrogel cushion was used to impose compressive stress to Lamin-SS expressing cells. Schematic representation of the experimental workflow (left). Representative confocal images of the Lamin-SS expressing cells before (magenta) and under the gel cushion (cyan) (middle) and the subsequent change in the nuclear morphology (right). Scale bar 10 μm. **f** riFRET efficiency images of Lamin-SS expressing cells before and under the gel cushion together with representative blow-up images. Scale bars, 10 μm. **g** Quantified riFRET (mean ± SEM) in Lamin-SS expressing cells subjected to compressive stress ($n$ = 164 cells, two biological replicates). Two-tailed Wilcoxon matched-pairs test (****$p$ < 0.0001).

reduces forces across cell–cell adhesions[32,33] and others have shown similarly reduced tension across nesprin-2[22]. Here, induction of epithelial to mesenchymal transition (EMT) by using TGF-β also resulted in decreased lamin A/C forces (Fig. 4b). Median sFRET efficiency of Lamin-SS was 16% (mean 0.17 ± 0.01 SEM) without TGF-β1 and 22% (mean 0.214 ± 0.004 SEM) after TGF-β1 treatment. The efficiencies were 40% (mean 0.38 ± 0.01 SEM) and 40% (mean 0.39 ± 0.01 SEM) with Lamin-TM, respectively. However, prior studies have suggested that chromatin decondensation increases during EMT[34]. Additionally, chromatin condensation changes have been shown to affect nuclear stiffness[35,36], and thus may also have the potential to regulate the mechanical state of the nuclear lamins. To directly modulate chromatin condensation, we used a histone deacetylase inhibitor (TSA) and a histone trimethyl demethylase inhibitor (methylstat) to decondense and condense chromatin, respectively. Cells treated with TSA to decondense chromatin had significantly decreased lamin A/C force

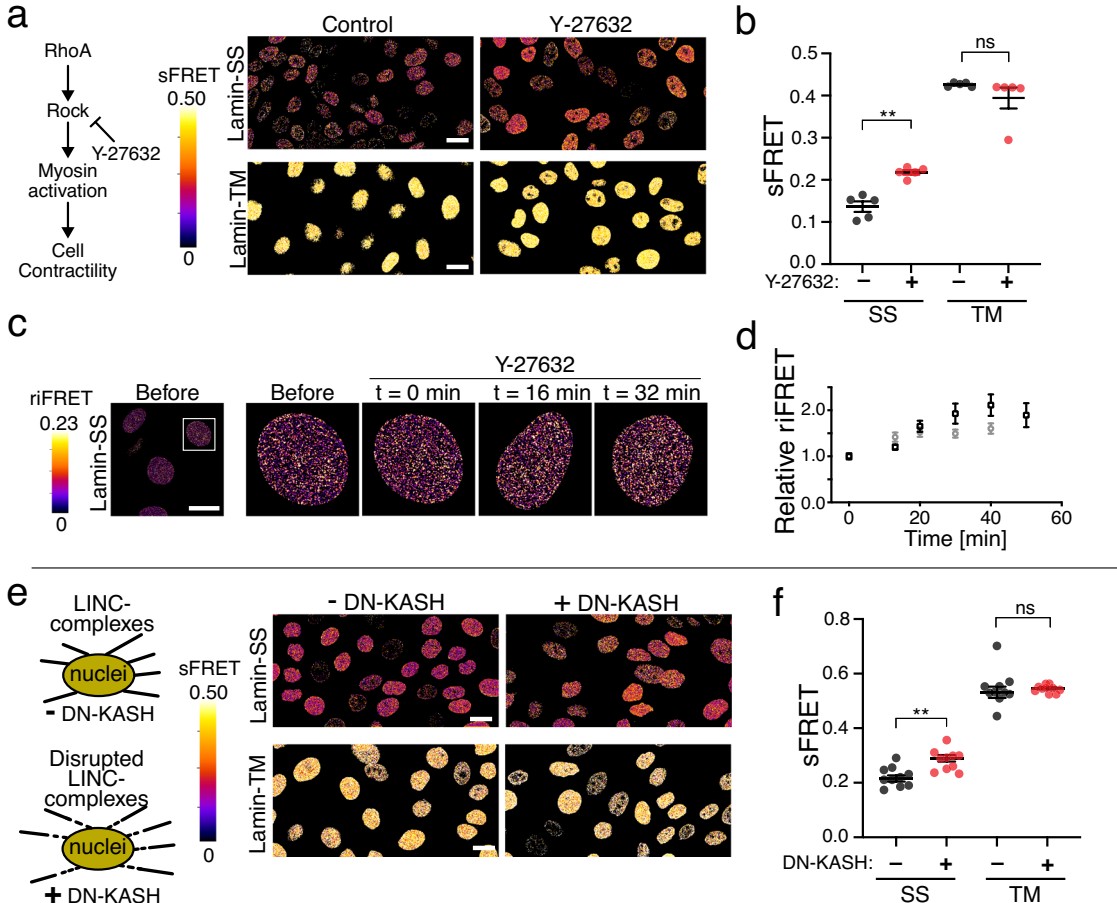

**Fig. 3 | The effect of actomyosin contractility, actin cytoskeleton integrity and LINC complexes on nuclear force transduction. a** sFRET efficiency images of Lamin-SS and Lamin-TM after cell contractility inhibition (Y-27632, 50 μM, 1 h). Scale bars, 20 μm. **b** Quantified sFRET (mean ± SEM) of Lamin-SS and Lamin-TM sensors after ROCK-inhibition (*n* = 5 fields, 3 biological replicates). Ordinary one-way ANOVA Tukey's multiple comparisons (for Lamin-SS (**) *p* = 0.005 and Lamin-TM (ns) *p* = 0.4132). **c** Lamin-SS riFRET imaging during ROCK-inhibition (Y-27632, 50 μM, added at time point 0 min). Scale bar 20 μm. **d** Quantified relative change

(mean ± SEM) in Lamin-SS riFRET ratio during ROCK-inhibition (*n* = 152 cells, two biological replicates, black and gray). **e** Disruption of LINC complexes by dominant-negative KASH (DN-KASH) expression (induction for 24 h). sFRET efficiency images of Lamin-SS and Lamin-TM after LINC disruption. Scale bars, 20 μm. **f** Quantified sFRET (mean ± SEM) of Lamin-SS and Lamin-TM after LINC complex disruption (*n* = 10 fields, two biological replicates). Ordinary one-way ANOVA Tukey's multiple comparisons (for Lamin-SS (**) *p* = 0.004 and Lamin-TM (ns) *p* = 0.1722).

(Fig. 4c). Median sFRET efficiency of Lamin-SS was 15% without TSA (mean 0.15 ± 0.004 SEM) and 20% after TSA treatment (mean 0.21 ± 0.01 SEM). The efficiencies were 36% (mean 0.361 ± 0.004 SEM) and 37% (mean 0.371 ± 0.004 SEM) with Lamin-TM, respectively. The treatment did not affect the nucleus circularity or the cross-section area (Supplementary Fig. 9). The increased histone acetylation after TSA treatment was confirmed by significantly higher immunostaining intensity of acetylated H3K27 (Supplementary Fig. 10). We also investigated the possibility that the lamin A/C organization is affected by the TSA and subsequently affects the Lamin-SS FRET. Here we used antibodies which recognize two different epitopes in the lamin A/C. The C-terminal epitope labeling has been shown to depend on higher organization of lamin A/C filaments[11] as the other antibody recognizes the rod-domain region of the lamin A/C (Fig. 4e). Thus, the labeling ratio of these antibodies can be used to probe the general level of nuclear lamina accessibility and organization. Ratiometric imaging by using the antibodies indicated that the TSA treatment did not alter the organization of A-type lamins (Fig. 4f). In control cells the antibody labeling ratio was 0.72 ± 0.30 and in the TSA-treated cells 0.69 ± 0.18. In addition, the treatment did not affect Lamin-SS binding to the nuclear lamina and the binding dynamics unaltered as shown by the FRAP experiments (Supplementary Fig. 11). In comparison to TSA induced decondensation of chromatin, cells treated with methylstat to

condense chromatin exhibited a small but non-significant increase in lamin A/C force (Fig. 4d). Median sFRET efficiency of Lamin-SS was 17% (mean 0.17 ± 0.01 SEM) without methylstat and 15% (mean 0.14 ± 0.01 SEM) after methylstat treatment. The efficiencies were 42% (mean 0.40 ± 0.02 SEM) and 40% (mean 0.39 ± 0.003 SEM) with Lamin-TM, respectively. Similarly, to TSA treatment, methylstat also did not affect A-type lamin organization (Fig. 4g), in control cells the antibody labeling ratio was 1.14 ± 0.13 and in the methylstat-treated cells 1.22 ± 0.33.

Overall, when summarizing the different conditions and treatments, the high osmolarity showed the biggest effect on Lamin-SS FRET, followed by the effect of reduced actomyosin contractility and chromatin compaction (Fig. 4h). Lower FRET was detected only when cells were mechanically compressed, which potentially led to stretching of the nuclear lamina.

## Nucleoplasmic lamins also experience mechanical force

Intriguingly, we detected similar levels of FRET at the nuclear perimeter and in the nucleoplasm (Fig. 1c, d). In addition, we quantified the Lamin-SS and Lamin-TM FRET in the nuclear rim and nucleoplasm by using FLIM (Fig. 5a). The data indicated that the lifetime and thus the FRET did not change between the nuclear perimeter and nucleoplasm (Fig. 5b). Together these data suggest that nucleoplasmic A-type

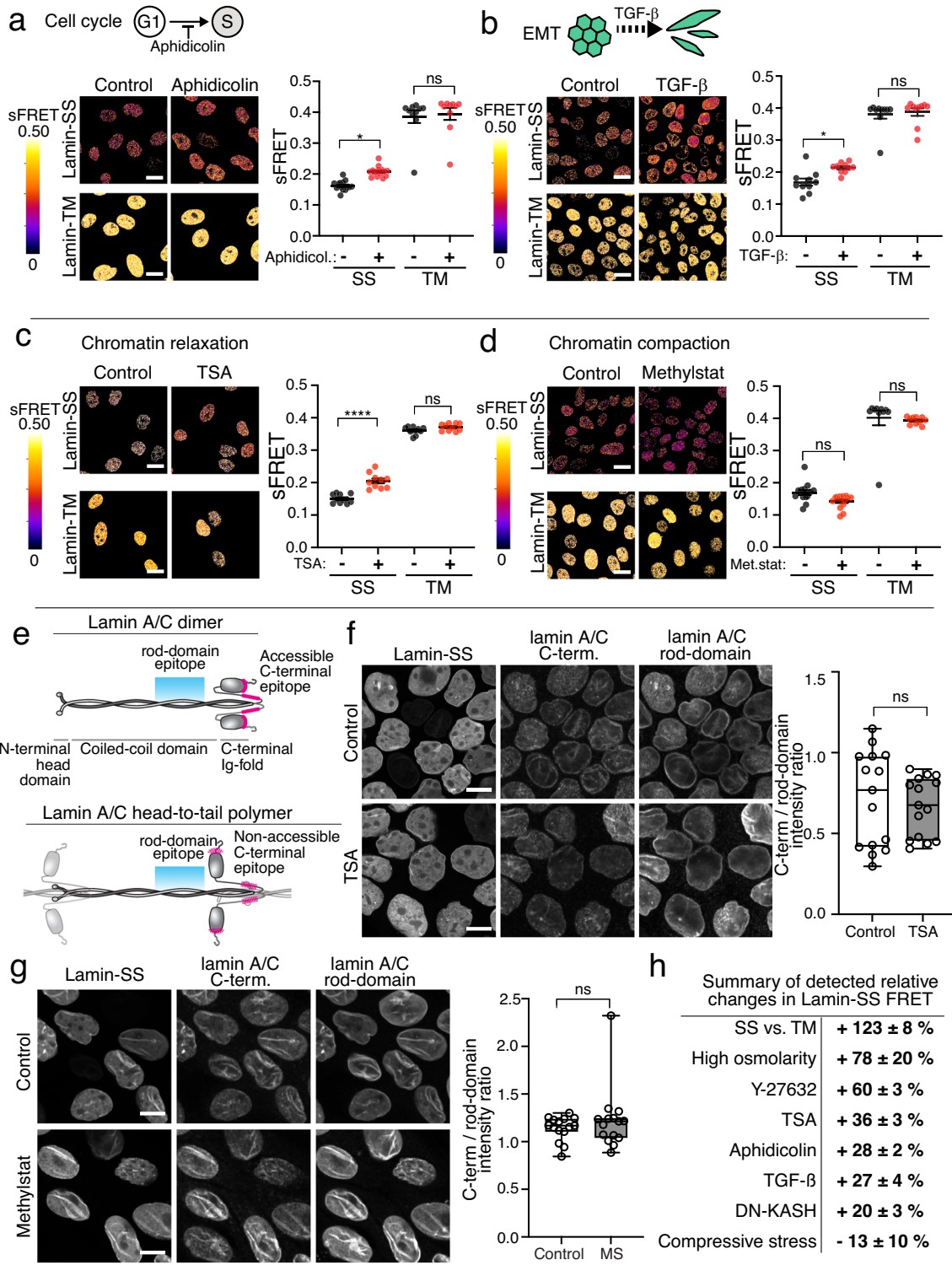

**h** Summary of detected relative changes in Lamin-SS FRET

| | |
|---|---|
| SS vs. TM | **+ 123 ± 8 %** |
| High osmolarity | **+ 78 ± 20 %** |
| Y-27632 | **+ 60 ± 3 %** |
| TSA | **+ 36 ± 3 %** |
| Aphidicolin | **+ 28 ± 2 %** |
| TGF-β | **+ 27 ± 4 %** |
| DN-KASH | **+ 20 ± 3 %** |
| Compressive stress | **- 13 ± 10 %** |

lamins also experience significant forces. Nucleoplasmic lamins have been shown to be assembled and to interact with chromatin[37]. The lamin proteins are known to bind with core histones via their C-terminal tails, thus allowing the force transduction between the lamins and chromatin[38,39]. To further examine nucleoplasmic lamins, as well as the potential for force transmission between nuclear lamins and chromatin, we developed an additional nanobody sensor, using a previously developed H2A-H2B nanobody[40], to measure mechanical tension between histone H2A-H2B and lamin A/C (Lamin-histone-SS; Fig. 5c). This sensor localized predominantly to the nucleoplasm and when compared to Lamin-SS was not similarly detected at the nuclear

rim. This was evident in the lamin A/C and H2A staining, where the sensor distribution was highly correlated to H2A (Fig. 5d). The Lamin-histone-SS exhibited reduced FRET in comparison to a truncated sensor Lamin-histone-TM (Fig. 5e). Lamin-histone-SS had a median sFRET efficiency of 27% (mean 0.27 ± 0.02 SEM), compared to Lamin-histone-TM with 43% (mean 0.41 ± 0.02 SEM). For further validation Lamin-histone-SS was examined in the MDCK LMNA knockout cell line (Fig. 5f). Lamin-histone-SS exhibited higher FRET in the knockout cell line, indicating that Lamin A/C proteins are necessary for tension across this sensor (Fig. 5f, g). Lamin-histone-SS median riFRET was 19% (mean 0.191 ± 0.006 SEM) in WT and 21% (mean 0.222 ± 0.006 SEM) in

**Fig. 4 | The effect of cell cycle, EMT and chromatin organization on lamin A/C strain. a** Analysis of Lamin-SS and Lamin-TM sFRET (mean ± SEM) after cell cycle synchronization to early S-phase (Aphidicolin, 3 μg/mL, 24 h) (n = 10-15 fields, three biological replicates). Ordinary one-way ANOVA Tukey's multiple comparisons (for Lamin-SS (*) p = 0.0229 and Lamin-TM (ns) p = 0.9619). **b**, Analysis of Lamin-SS and Lamin-TM sFRET (mean ± SEM) after EMT induction by growth factor treatment (TGF-β1, 2 ng/mL, 24 h) (n = 10 fields, two biological replicates). Ordinary one-way ANOVA Tukey's multiple comparisons (for Lamin-SS (*) p = 0.0311 and Lamin-TM (ns) p = 0.9666). **c** Analysis of Lamin-SS and Lamin-TM efficiency (mean ± SEM) after treatment by histone deacetylase inhibitor (TSA, 200 nM, 4 h; n = 10 fields, three biological replicates). Ordinary one-way ANOVA Tukey's multiple comparisons (for Lamin-SS (****) p < 0.0001 and Lamin-TM (ns) p = 0.4804. **d** Analysis of Lamin-SS and Lamin-TM (mean ± SEM) efficiency after treatment by histone demethylase inhibitor (methylstat, 2.5 μM, 48 h) of the cells (n = 10–15 fields, two biological replicates). Ordinary one-way ANOVA Tukey's multiple comparisons (for Lamin-SS (ns) p = 0.2282 and Lamin-TM (ns) p = 0.9584). **e** Localization of epitopes for the used lamin A/C rod-domain and C-terminal antibodies. The C-terminal epitope accessibility depends on lamin filament organization. **f** Analysis of nuclear lamina organization in TSA-treated cells. Confocal microscopy maximum intensity projections (CM-MIP) of control (upper panels) and TSA-treated (600 nM, 4 h, lower panels) Lamin-SS expressing cells, immunolabeled against lamin A/C. Quantified fluorescence intensity ratio of lamin A/C labeling in control and TSA-treated cells (box from 25th to 75th percentile, median, whiskers from min to max, n = 15 fields, three biological replicates). Unpaired two-tailed Student's t test ((ns) p = 0.6, t = 0.5, df=28). **g** Analysis of lamina organization in methylstat-treated cells. CM-MIP of control (upper panels) and methylstat-treated (2.5 μM, 48 h, lower panels) Lamin-SS expressing cells, immunolabeled against lamin A/C. Quantified fluorescence intensity ratio of lamin A/C labeling (box from 25th to 75th percentile, median, whiskers form min to max, n = 15 fields, 3 biological replicates). Unpaired two-tailed Student's t test ((ns) p = 0.4, t = 0.9, df = 28). **h** Summary table of the relative changes in the quantified FRET changes in different conditions. Scale bars, **a–d** 20 μm and **f–g** 10 μm.

LMNA KO cells. In comparison, the Lamin-histone-TM median riFRET was 27% (mean 0.261 ± 0.004 SEM) in WT cells and 28% (mean 0.272 ± 0.003 SEM) in LMNA KO cells. The overall higher riFRET of the Lamin-histone-TM in comparison to Lamin-histone-SS in KO cells can be explained by the possible increase in the intermolecular FRET. Lleres et al. were able to use FRET between histone fluorescent proteins to measure the compaction state of the chromatin[41]. Thus, in the case of Lamin-histone-TM the higher riFRET might rise from sensor-to-sensor FRET.

Although lamins and histones are known to physically associate[38], the Lamin-histone-SS establishes that mechanical forces can be transduced between chromatin and A-type lamins. Taken together, these results with both the Lamin-SS and Lamin-histone-SS indicate that also nucleoplasmic lamins experience and can transmit significant levels of mechanical force.

## Discussion

Forces and mechanical stresses at the cellular and protein levels are difficult to quantify directly. Although a number of approaches have used externally applied force to study mechanotransmission of force onto and within the nucleus[2,28], more recently we and others have developed force-sensitive biosensors for nesprin proteins[21–23], as well as identified force-sensitive antibodies[11], which are technical approaches that can be used to characterize nuclear forces without the need for externally applied forces. In this work we established a nanobody-based FRET-strain sensor concept, which enabled mechanical studies of the protein dense and filamentous nuclear lamina. The sensor binds to A-type lamins and the measured FRET value depends on the strain state of the sensor. We employed 3 different imaging and analysis approaches to quantify the changes of FRET to show the robustness of the FRET approach. We used FLIM to follow the FRET independent of the intensity. In addition, we employed two intensity-based FRET quantification methods (spectral imaging (sFRET) and ratiometric imaging (riFRET)) to measure the FRET by common confocal microscopy. Using this biosensor, we demonstrate the lamin A/C network experiences significant mechanical force, which is affected by actin, myosin, and nuclear-cytoskeletal connections (Fig. 2). Additionally, we show that there exists a positive correlation between chromatin condensation and lamin A/C forces (Fig. 3). Intriguingly the nucleoplasmic lamin A/C forces were of a similar level to the forces of lamin A/C at the NE (Fig. 4). Our data shows that nuclear lamina is an interface experiencing forces rising from inside (chromatin) and outside (cytoskeleton, nucleus deformation) of the nucleus (Fig. 4h).

There are many prior studies demonstrating that the nucleus can experience and sustain mechanical loading. Physical perturbations of cells and isolated nuclei have shown that nuclear lamins deform under mechanical loading[42,43]. Nuclear lamina is also a mechanosensitive structure and its organization changes according to cellular tensile state or cell substrate rigidity[10,11]. Furthermore, forces are transduced through the nuclear lamina and can affect chromatin organization[2]. Mechanical integrity of the nucleus has also been shown to depend on A-type lamins[44]. In line with this, loss and mutant forms of lamin A/C lead to force-induced nuclear rupture and DNA damage[43,45,46]. These studies indicate that nuclear lamina experiences and responses to mechanical cues. Our newly developed strain sensor adds to this prior work by demonstrating that the lamin A/C network is subject to constitutive mechanical tension in adherent cells. This kind of direct measurement of tensile state or mechanical strain of the lamina has not been previously possible.

In addition to localizing to the nuclear periphery (at the NE), A-type lamins are present throughout the nucleoplasm[5]. Work by Roland Foisner and colleagues have shown that nucleoplasmic lamins associate with LAP2α and may have important roles in regulation of chromatin[7]. Loss of LAP2α has been shown to decrease nucleoplasmic lamin mobility, resulting in a potentially more assembled filamentous nucleoplasmic lamin structure and decreased chromatin motion[37]. Our observation of significant mechanical forces on nucleoplasmic A-type lamins (Fig. 5) further supports the idea that together with chromatin these lamins provide important structural and mechanical properties to the nucleus[31]. Additionally, our histone-lamin sensor shows that nucleoplasmic lamins are an important component mechanically interacting with chromatin (Fig. 5). However, due to the abundance of the histone proteins in the nuclear interior, our Lamin-histone sensor is strongly localized to the chromatin. Therefore, with current microscopy resolution we cannot directly compare the lamin - histone force transduction in the vicinity of the nuclear lamina and in the nuclear interior.

Our FRAP experiments demonstrated simultaneous, dual binding of both nanobodies in Lamin-SS to lamin A/C filaments (Fig. 1). Lamin protein dimers form partly staggered head-to-tail polymers with repetitive features every 40 nm[6]. Lamin filaments are assembled from two laterally interacting head-to-tail polymers and show 20 nm periodicity (Supplementary Fig. 3) and filament length of approximately 400 nm within the nuclear lamina[6]. Since the nanobody recognizes a single epitope in the end of the coiled-coil domain and the estimated Lamin-SS length is 40 nm (when fully extended), it is possible that each nanobody in the sensor is binding to two lamin A/C proteins in a single filament (Supplementary Fig. 1S3). Mechanically lamin filaments are flexible and their persistence length is 50–2700 nm[6]. Therefore, in the length scale of the Lamin-SS sensor (40 nm), the lamin filament is essentially a straight rod. Thus, it is unlikely that the bending or straightening of lamin filament would cause substantial strain in the sensor. However, recently Medalia group proposed a model based on thorough characterization of nuclear lamina mechanics, that under low force lamin filament length could increase via unfolding or sliding of lamin proteins within the filament[9]. This could generate strain also

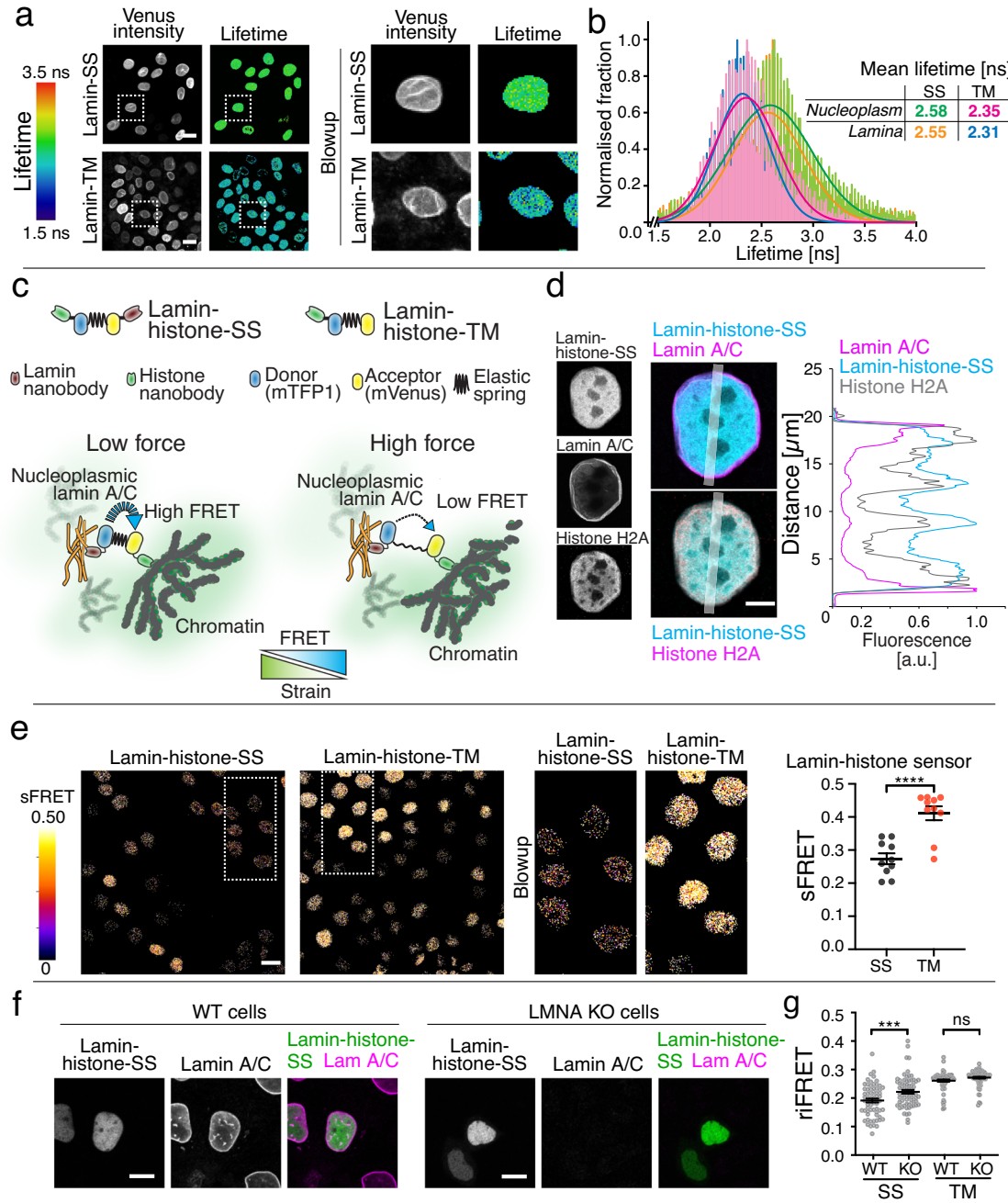

**Fig. 5 | Mechanical strain in nucleoplasmic lamin A/C filaments.**
**a** Representative acceptor (venus) intensity image together with donor (mTFP1) fluorescence lifetime microscopy images of Lamin-SS and Lamin-TM expressing cells. Scale bars, 20 μm. Blowup images show equal distribution of lifetimes throughout the nucleus. **b** Donor fluorescence lifetime histograms show highly similar lifetimes and thus FRET for nuclear rim and nucleoplasm, indicating similar strain in lamin A/C in the nuclear lamina and in the nuclear interior (*n* = 26 and *n* = 26 cells for Lamin-SS and Lamin-TM, respectively, two biological replicates). **c** Schematic representation of the FRET based lamin A/C - histone H2A strain sensor (Lamin-histone-SS), truncated control sensor (Lamin-histone-TM), and the working mechanism of the force sensing between lamins and chromatin. **d** Representative confocal Airyscan xy-sections of immunolabeled lamin A/C (top, magenta), histone

H2A (magenta, bottom) and the expressed Lamin-histone-SS sensor (cyan) along with corresponding fluorescence line-profiles (*n* = 5 and *n* = 4 cells, respectively, two biological replicates). Scale bar 5 μm. **e** sFRET efficiency images and quantified sFRET efficiency (mean ± SEM) of Lamin-histone-SS and Lamin-histone-TM sensors (*n* = 10 fields, two biological replicates). Scale bar 20 μm. Unpaired two-tailed Student's *t* test ((****) *p* < 0.0001, *t* = 5.14, df = 18). **f** WT and LMNA KO cells transiently transfected with Lamin-histone-SS. Scale bars 10 μm. **g** Quantified riFRET (mean ± SEM) of Lamin-histone-SS (*n* = 67 and *n* = 72 cells, respectively, two biological replicates) and Lamin-histone-TM (*n* = 67 and *n* = 78 cells, 2 biological replicates) in WT and LMNA KO cells. Ordinary one-way ANOVA Tukey's multiple comparisons (for Lamin-histone-SS (***) *p* = 0.0003 and Lamin-histone-TM (ns) *p* = 0.4756).

to the dual bound Lamin-SS sensor. Alternatively, our model indicates that within the dense lamina meshwork, the Lamin-SS can bridge two adjacent filaments (Supplementary Fig. 3). Therefore, shear and reorganization of the lamin network can also lead to changes in the detected Lamin-SS FRET. Thus, either single filament or dual filament

binding scenarios are possible. However, the detected FRET efficiencies of the Lamin-SS were low already in control conditions, indicating considerable strain in the TSmod sensor module. This most likely highlights the epitope-to-epitope distance in lamina, which leads to a strained sensor and reduced FRET.

A further hindrance in our detailed understanding of how Lamin-SS is interacting with the Lamin A/C network is that the epitope of the lamin A/C nanobody has not yet been identified. We also note that we previously reported that mechanical forces can regulate lamin A/C epitope accessibility[11]. Although our control experiments (Fig. 3) did not show major changes in accessibility of other lamin A/C epitopes with TSA and methylstat treatments, it may be possible that the Lamin-SS epitopes are also influenced by the physical arrangements of the nuclear lamina, in addition to possible force induced direct lamin unfolding. However, our FRAP data indicates that TSA treatment does not influence Lamin-SS binding to the lamina (Supplementary Fig. 11), further arguing against force-induced changes in binding of Lamin-SS.

We also considered the possibility that changes in lamin A/C structure could affect the density of the sensor. Extremely close packing of multiple sensors can lead to intermolecular FRET, FRET occurring between neighboring sensors when sensors are closely packed together[47]. We note that the truncated Lamin-TM did not exhibit FRET changes in response to experimental treatments (Figs. 2f, j, 3a–d, and 4f). Because Lamin-TM should also be subjected to similar changes in sensor densities and packing as Lamin-SS during these treatments, we assume that treatment-induced changes in Lamin-SS FRET are occurring not from changes in intermolecular FRET, but instead changes in tensile forces across a single sensor.

Through our FRAP analysis of binding, we observed that the turnover rates of the Lamin-SS (Fig. 1) are much faster than the longer turnover rates of Lamin A[26,48]. Thus, the reduced FRET of Lamin-SS, which has a dual-binding time of approximately 30 seconds, is due to dynamics of the lamin network and the turnover of lamins is not affecting the process. However, in the nucleoplasm we can't exclude the possibility that chromatin diffusion creates additional strain to the Lamin-histone-SS during the dual-binding of the sensor. Our FRAP analysis indicates only 10% of Lamin-SS exists with both nanobodies simultaneously bound (Fig. 1g), which is surprising given the large decrease in FRET observed with this sensor as compared to Lamin-TM. One possibility is there is a delay in the relaxation of the sensor after unbinding. While Lamin-SS sensors may bind, unbind, and re-bind rapidly, the extension of the flagelliform peptide may persist over a longer timescale, remaining extended even when the sensor is in a single or unbound state. Similarly, it is possible that mechanical forces applied across the sensor causes one or more of the fluorescent proteins to partially unfold[49]. In addition, it has been reported that even before complete unfolding of the fluorescent protein, the brightness can be substantially reduced by force induced mechanical quenching[50]. This would result in decreased fluorescence of the protein[51] and therefore changes in the detected FRET. If true, the refolding of the fluorescent proteins would likely occur on a longer timescale, resulting in altered FRET persisting in the single or unbound states. If there is delayed refolding or relaxation of the sensor this raises the possibility that there would exist a temporal lag in Lamin-SS detection of force relaxation. We note that the sensor was readily able to detect lamin A/C relaxation after 15 min of osmotic nuclear shrinkage (Fig. 2c) and ~20 min of Y-27632 treatment (Fig. 2g), indicating that any temporal lag is modest and that force-induced changes in Lamin-SS are reversible.

While our approach might be considered to be the first use of TSmod in an intermolecular strain/force sensor (between two proteins), we note the prior work of Alex Dunn and colleagues in which TSmod was also used similarly in an intermolecular manner to measure forces exerted between integrins and the extracellular matrix[52]. An important benefit to our nanobody-based intermolecular sensor is that it remains completely genetically encodable, allowing for cellular expression of the sensor. In addition, the nanobody-based sensor does not require construction of fusion proteins with internally inserted TSmod, which can be challenging due to the size of the TSmod. Furthermore, the implementation of nanobodies with TSmod enables this approach to be adapted to other proteins for which there are existing nanobodies, such as actin[53] and vimentin[54]. Finally, existing nanobody epitope tags (C-Tag, Spot-tag, and ALFA-tag)[55], as well as anti-GFP and anti-RFP nanobodies[56] can be used in the development of generalized nanobody strain sensors for virtually all proteins.

This technical advancement provides significant insight into nuclear mechanics, by providing the first direct protein-level measurements of nuclear lamina strain. Lamin A/C strain, presumably the result of tensile and compressive mechanical forces, is dynamic and influenced in both an outside-in (actomyosin, LINC complex) and inside-out (chromatin) manner. Additionally, we show that intra-nuclear lamins also experience significant levels of strain, providing additional evidence that nucleoplasmic lamins are an important structural element of the nucleus. This work demonstrates the potential for nanobody-based biosensors to be further utilized to measure mechanical forces between homotypic and heterotypic protein associations.

## Methods

### Sensor design

The sensor to measure mechanical forces on the nuclear lamins was designed using an existing lamin A nanobody. The nanobody was previously developed by Rothbauer et al.[24] and the expression vector coding for the lamin-nanobody was provided by ChromoTek & Proteintech (Planegg-Martinsried, Germany) with a corresponding end user license agreement. The sensor is designed such that an existing FRET-force biosensor, known as TSmod[13] is flanked on either side by the lamin A nanobody $V_HH$ sequence (Fig. 1a). To ensure nuclear localization of the protein a c-myc NLS was inserted between each nanobody and TSmod. Additionally, the C-terminal lamin A nanobody was designed by using the reverse sequence of the $V_{HH}$ for orientation of the nanobody outwards from TSmod. The entire sequence of the sensor was synthetically cloned by GeneArt (Thermo Fisher Scientific) into pcDNA 3.3.

A second sensor was developed designed to measure forces between the nuclear lamina and histones. This sensor consists of a nanobody which binds to the Histone H2A-H2B heterodimer that was previously developed by Jullien et al.[40] and is also commercially distributed as a "chromobody" by Chromotek. The sensor was designed similarly to the lamin sensor, with the N-terminal lamin A sensor being replaced with the histone nanobody (histone nanobody-TSmod-reverse lamin A nanobody).

### Simulation of the sensor size

The model for Lamin-TS was constructed with PyMOL[57] from PDB structures 2HQK (mTFP1), 1MYW (Venus), and 4JVP (structure of anti-Hepatitis C virus nanobody). The linker regions connecting the existing PDB models were created with the PyMOL builder tool as an anti-parallel beta-sheet to mimic a linear molecule structure. Structures were joined together and aligned with each other using editing tools in PyMOL.

For the folded version of the lamin-TS, the linker regions of the construct were folded in 50 ns long MD simulation. The simulation was run with GROMACS program version 2021.4[58]. Amber ff14SB force field[59] with spc/e water model in 0.15 KCl solution were used[60], and the total charge of the system was set to neutral. System energy minimization was done with a steep integrator. All of the following simulations were performed with 2 fs timestep. System was equilibrated first in NVT simulation of 1 ns with V-rescale[61] coupling at 300 K, and then in NPT simulation of 1 ns with V-rescale coupling at 300 K and Berendsen[62] coupling at 1 bar. Final folding simulation of 50 ns was done with V-rescale coupling at 300 K and Berendsen coupling at 1 bar, with position restraints only allowing Z-directional movement in the ends of the linker chains to better mimic the linker folding when coupled to large proteins.

The conformations with the best terminal alignment and most compact structure were selected to best simulate the folded state of the Lamin-TS. Simulated linker regions were added back to the model to replace the open versions of the regions and create a representation of the folded version of the structure.

The PDB model was created to evaluate the possible length of the open and folded forms of the linker, and most probably does not represent the true physiological conformation of the sensor but provides an insight to the possible lengths of the open and folded conformations of the sensor. Minor differences in the sequences of the models used in the modeling should provide no meaningful differences in the lengths.

### Modeling of sensor binding to the nuclear lamina

The data from Turgay et al.[6] was used as a model of the nuclear lamina organization and lamin filament assembly. Two subvolumes (from Fig. 2 of Trugay et al.) were imported to ImageJ and pixel size was scaled accordingly to the scale bar of the image. Next the images were converted to 8-bit gray scale images, Gaussian smoothed (radius 2) and manually segmented. Next the data was skeletonized by using the *skeletonize* function of ImageJ. This skeletonized data was then overlaid with the previous 8-bit gray scale images of the lamina. The sensor radius was then drawn to the image using Adobe Photoshop (v. 22.3 or higher). The schematic representation of the lamin filament assembly was drawn in Adobe Illustrator according to the data published in the supplementary file of Turgay et al.[6].

### Cells

MDCK II (obtained from Jennifer Lippincott-Schwartz lab) was used for all experiments, except MDCK II (obtained from Aki Manninen lab (University of Oulu, Finland), originally from ECACC #00062107) was used for lamin A knockout experiments. The cells were maintained in high glucose DMEM (Thermo Fisher Scientific) supplemented with 10% fetal bovine serum (Thermo Fisher Scientific) and 1% penicillin/streptomycin (Thermo Fisher Scientific) under standard cell culture conditions. To generate stable cell lines, MDCKs were transfected with the TSmod and selected using G418. For the DN-KASH experiments, DN-KASH inducible Lamin-SS cells and DN KASH inducible Lamin-TM cells were made into stable cell lines. To generate a system for doxycycline-inducible DN-KASH Lamin-SS cells and doxycycline-inducible DN-KASH Lamin-TM cells, the previously established doxycycline-inducible DN-KASH MDCK cells[63] were electroporated with Lamin-SS pcDNA and Lamin-SST pcDNA separately. Cells expressing both DN-KASH and Lamin-SS/TM were extracted with cloning rings and were clonally expanded.

### Establishment of LMNA knockout with CRISPR/Cas9

To generate a pre-LMNA knockout (KO) MDCK II cell line with CRISPR/Cas9, single guide RNAs (sgRNAs) were custom made from Invitrogen backbone from their LentiArray™ Human CRISPR Library and designed against LMNA1 gene N-terminus in CanFam 3.1 reference genome (https://www.ncbi.nlm.nih.gov/nuccore/NM_001287151.1, GeneID: 480124) with an online guide design tool. LMNA target sequence: atggagac cccgtcccag cggcgcgcca cccgtagcgg ggcgcaggcc agctccaccc cgctgtcgcc cacccgcatc acccggctgc aggagaagga ggacctgcag gagctcaatg accgcctggc ggtctacatc gaccgtgtgc gctctctgga gacggagaac gcggggctgc gccttcgcat caccgagtcg. The sgRNA_LMNA_N1 nucleotide sequence was CACGGTCGATGTAGACCGCC (on-target locus chr7:−41719582). For expression, the sgRNA_LMNA_N1 (300 ng) and pCDNA3.1-dCas9-2xNLS-EGFP (gift from Eugene Yeo, Addgene #74710) were transfected by using the Neon™ electroporation system (1650 V, 20 ms, 1 pulse; Thermo Fisher Scientific) followed by selection of GFP-positive cells with G418 (0.75 mg/mL, Merck) and FACS sorting (BD FACSAria Fusion, BD Biosciences)[64]. The KO cells was verified via immunoblotting, immunostainings and further by sequencing to specifically detect the knockout region.

### Immunoblotting

For immunoblotting, MDCK II and MDCK LMNA KO cells were washed with PBS and lysed in buffer containing 50 mM Tris-Cl pH 7.5, 1% Triton X-100, 1 mM EDTA, 150 mM NaCl, 50 mM NaF, 10% glycerol, 1 mM phenylmethanesulfonyl fluoride, 8.3 μg/ml aprotinin, 2 mM vanadate, and 4.2 μg/ml pepstatin. The lysates were centrifuged and used for SDS-PAGE with Mini-PROTEAN® TGX™ Precast gel (Bio-rad Finland OY). Following transfer to Amersham™ Protran® nitrocellulose blotting membrane, the immunoblots were blocked with 2% BSA and incubated with primary antibodies: Anti-Lamin A/C antibody (E-1) (1:400, sc-376248, Santa Cruz Biotechnology) and Anti-Actin antibody, clone C4 (1:2000, MAB1501R, Merk-Millipore). The primary antibodies were detected using a mixture of goat-anti-mouse (DyLight 800, 1:5000, SA5-10176, Thermo Fisher Scientific) and goat-anti-rabbit (DyLight 680, 1:5000, 35568, Thermo Fisher Scientific) secondary antibodies and the signal was read using Odyssey CLx (LI-COR).

### Drug treatments

For actin depolymerization studies, Cytochalasin D (cat # 11330, Caymen Chemical) was used at 10 μg/mL for 1 h. To inhibit Rho A kinase, 50 μM Y-27632 (cat #72302, Stem Cell Technologies) was used for 1 h prior to FRET imaging to reduce myosin activity. For EMT induction, recombinant human TGF-β1 (R&D systems) was used to induce EMT at a concentration of 2 ng/mL for 24 h. Modifications in DNA ultrastructure were done to condense or decondense chromatin with the use of 600 nM trichostatin A (TSA) for 4 h (Cayman Chemical Company), to increase euchromatin, and 2.5 μM methylstat (Sigma Aldrich) for 48 h, to increase heterochromatin. For the cell cycle synchronization assay, Aphidicolin (cat #57-361, Thermo Fisher Scientific) was used to block the cells in early S-phase, at 3 μg/mL for 24 h.

### Confocal microscopy of histones and nuclear lamina organization

Confluent non-treated or either trichostatin A (TSA, 4 h, 600 nM) or methylstat (48 h, 2.5 μM)-treated MDCK II wt, MDCK II-TS or MDCK II-truncated mutant cells were analyzed to ensure the drug treatments used in FRET-experiments did not affect the nuclear lamina organization. To detect changes in nuclear lamina organization, ratiometric fluorescence immunoassay was performed on MDCK II wt, MDCK II cells stably expressing Lamin-SS or MDCK II cells stably expressing Lamin-TM immunostained against either lamin A/C N-terminus, LA/C−N (validated in LMNA KO cells (see supplementary fig. S4), 1:500, mouse, clone: E1, #sc-376248, Santa Cruz Biotechnology, Texas, USA) and histone H3 lysine 27 acetylation, H3K27ac (ChIP-grade, 1:500, rabbit, #ab4729, Abcam, Cambridge, UK), or lamin A/C C-terminal, LAC/C−C (validated in LMNA KO cells, 1:400, mouse, clone: 131C3, #ab8984, Abcam) or lamin A/C rod domain, LA/C-rod (KO-validated recombinant rabbit-monoclonal, 1:50, clone: EP4520-16, #ab133256, Abcam). The used secondary antibodies were goat anti-mouse Alexa 488 (1:200, #A-11001, Thermo Fisher Scientific), goat anti-mouse 568 (1:200, #A-11004, Thermo Fisher Scientific), goat anti-mouse Alexa 647 (1:200, #A-21235, Thermo Fisher Scientific), goat anti-rabbit Alexa 568 (1:200, #A-11011, Thermo Fisher Scientific) and goat anti-rabbit Alexa 647 (1:200, # A-21245, Thermo Fisher Scientific). In the ratiometric analysis, the mean nuclear intensity of lamins was quantified from confocal microscopy images acquired with identical imaging settings, followed by calculating the respective intensity ratios. Imaging was done on a Nikon A1R+ laser scanning confocal microscope (NIS Elements, v. 5.11) mounted in Nikon Eclipse Ti2-E (Nikon Instruments, Tokyo, Japan). Nikon 60X/1.40 Apo DIC N2 oil immersion objective was used in the experiments. Solid state lasers with excitation wavelengths

488 nm, 561 nm, and 640 nm were used in excitation. The emissions were collected with 525/50, 540/30, and 595/50 bandpass filters, respectively. The laser intensities were adjusted to avoid photobleaching and the detector sensitivity was adjusted to optimize the image brightness and to avoid saturation. Laser powers and detector voltages were determined individually per treated antibody pair, and after the initial setting kept constant for each sample to allow ratiometric imaging and quantitative comparison of the fluorescence intensities within the drug-treated and non-treated control samples. The images were 1024 × 1024 pixels and the pixel size was 103.6 μm in x/y. The images were acquired without averaging and by first focusing on the bottom surface of the sample, where the position of the sample stage was set as z0 = 0. The fluorescence signal intensities from all emission channels were then collected from bottom to top as optical z-series with 200 nm step size. The pinhole was set to 0.9 (physical pinhole size 34.76 μm). The analysis was done in ImageJ software by making maximum intensity projections from the acquired z-stacks, and by using the LA/C-rod channel to segment the nuclei which was then used as a mask to measure the maximum signal intensities for all channels. The mean intensities of the nuclei, the background, and the total images were determined. To detect changes in the lamin organization the nuclear lamin intensity ratio (LA/C–C:LA/C-rod) was calculated from the nuclear intensities of which the background was subtracted.

### Super-resolution airy-scan imaging

Zeiss LSM 980 laser scanning confocal microscope (Zen Blue, v. 3.3.89.0007) with airyscan was used for fixed-cell experiments. The system was mounted on Axio Observer.Z1 microscope body and Plan-Apochromat ×63/1.4 oil immersion objective was used in the imaging. The used primary antibodies were anti-lamin A/C (ChIP-grade, 1:100, mouse, clone: 636, #sc-7292, Santa Cruz Biotechnology) and anti-histone 2A (ChIP-grade, recombinant rabbit-monoclonal, 1:200, clone: D6O3A, #12349 S, Cell Signaling). The used secondary antibodies were goat anti-rabbit Alexa 568 (1:200, #A-11011, Thermo Scientific), goat anti-mouse Alexa 647 (1:200, #A-21235, Thermo Scientific). The sensor, the immunolabelled lamin A/C, and histones were excited with 488 nm and 639 nm lasers using MDS488/561/639 triple dicroic and the emission was collected with band-pass 495-560 nm and long-pass 650 nm filters. The image size was set to 1032 × 1032 pixels, with pixel size of 43 nm and optical section collected with 170 nm intervals. Scanning was bidirectional with 2 μs pixel dwell time and averaging of 4 was used. Data was analyzed with ImageJ Fiji -distribution.

### Fluorescence recovery after photobleaching-experiments

Zeiss LMS780 laser scanning confocal microscope (ZEN Black, v. 2012 SP5) in inverted Cell observer microscope body was used in the experiments. MDCK cells stably expressing Lamin-SS or Lamin-TM or MDCK LMNA KO cells transiently expressing Lamin-SS or EGFP-lamin A were seeded on collagen-I -coated (50 μg/mL in PBS, 45 min in RT) high performance coverslips (Zeiss, #474030-9020-000) 1 d before the experiments. Prior imaging, the coverslips were mounted on imaging chamber (Aireka Cells, #SC15022, Aireka Scientific, HK, China) and placed in the microscope incubator (37 °C, 5% CO₂). Imaging was conducted by using 63X/1.2 WI C-Apochormat objective. Lamin-SS or Lamin-TM were excited with 514 nm laser line, pixel size was adjusted to 0.13 μm (zoom setting 4) and 256 × 256-pixel images were captured without averaging (195 ms scanning time per frame). In the FRAP experiment, images were collected with 250 ms intervals (249 images in total), and a bleaching was conducted after 9 scans. In the bleaching phase, a pre-drawn rectangular area of 75 × 10 pixels in the nuclear lamina was scanned 25 times (iterations) with 100% light intensity from 514 nm laser. The recovery was then followed for 240 frames.

### FRAP data analysis and simulations

FRAP recovery curves were measured by using ImageJ Fiji-distribution[65]. The drift of the nucleus during the imaging was corrected by using StackReg-plugin[66]. Next the fluorescence was measured from the lamina and from the whole nucleus. The data was then normalized in Microsoft Excel for Mac (version 16.55) according to Phair & Misteli[67]:

$$I(t) = \frac{\frac{lamina(t)}{lamina(t=0)}}{\frac{nucleus(t)}{nucleus(t=0)}} \tag{1}$$

Where lamina(t) is fluorescence in the lamina at time point $t$, lamina($t = 0$) is fluorescence in the lamina before the bleach phase, nucleus($t$) is the fluorescence of the whole nucleus at time point t and nucleus($t = 0$) is the fluorescence of the nucleus before the the bleach phase. Finally, the normalized recoveries were averaged.

Virtual Cell software[68,69] was used to simulate the FRAP experiment and fluorescence recovery. The model contains a free Lamin-SS sensor which can bind to an immobile binding site in the lamina (single bound sensor), this binding can then lead into release of the sensor or tighter binding, simulating the situation where the sensor is engaged from both nanobodies (dual bound sensor). The release of the dual bound sensor was assumed to happen via single bound-state. The Lamin-TM sensor behavior was assumed to behave otherwise similarly, only the dual binding opportunity was missing. The reaction network schematic is visualized in Supplementary Fig. 2. The Virtual Cell Models, "Lamin-SS_dual_binding" and "Lamin-TM_single_binding" by user "teihalai", can be freely accessed within the VCell software (available at https://vcell.org).

### SensorFRET imaging and analysis

Live cells were seeded on glass-bottom slides coated with 20 μg/mL fibronectin. DMEM was replaced with live cell imaging solution (cat #: A14291DJ, Thermo Fisher) supplemented with 10% FBS. Images were acquired using an inverted Zeiss LSM 710 confocal microscope (Zen Black, v. 2011 SP7) using both 405 nm or 458 nm excitation wavelengths from an argon laser source (Zeiss, Oberkochen, Germany). A 40x water immersion objective lens (NA = 1.1) was used for all imaging. Live cells were imaged in spectral mode using a 32-channel spectral META detector to record spectra of each pixel spanning wavelengths from 416 to 718 nm (with 9.7 nm spectral steps). Images were captured in 16-bit mode, scanned bi-directionally, and averaged 4 times. For sensorFRET based efficiency imaging, spectral images at both 405 and 458 nm excitation wavelengths were acquired. The normalized emission shape of the mTFP and mVenus fluorophores as well as the calibration parameter c (= 0.101) required for the sensorFRET analysis were experimentally determined from control cells expressing single fluorophores[70]. Intensity images were further processed and analyzed using a custom Python code, which involves background subtraction and removal of saturated pixels. For each data set, the data was acquired for at least 5 images per condition per experiment. Images were masked manually on ImageJ Fiji-distribution.

### Ratiometric riFRET measurements and analysis

Ratiometric riFRET imaging was used for FRET measurements involving paired FRET samples. Cell seeding and mounting was performed with similar protocol as in FRAP experiments. For live cell imaging, cells were placed in the microscope incubator (37 °C, 5% CO₂). Zeiss LSM 780 laser scanning confocal microscope (ZEN Black, v. 2012 SP5) equipped with Plan Apochromat ×63/1.4 oil immersion objective was used for the ratiometric FRET approach. FRET imaging and analysis was done by riFRET method described previously[71]. Briefly, the donor and acceptor were excited with a 458 nm line and a 514 nm line, respectively, from a multiline argon laser. The resulting fluorescence

was acquired between 465–500 nm for donor emission and 535–650 nm for acceptor emission with a 32-channel QUASAR GaAsP PMT array detector. FRET channel emission was obtained with donor excitation (458 nm) and detected through the acceptor emission channel. Cells stably expressing either donor or acceptor probes alone was used to determine the spectral croTMalk. riFRET plugin[12] for ImageJ was used for croTMalk correction of each channel and to calculate pixel by pixel-based riFRET efficiency. The riFRET efficiency from individual cells prior to and after treatment was used for analysis.

### Fluorescence lifetime imaging microscopy-FRET analysis
For FLIM, cells cultured in coverslips were fixed with 4% PFA for 10 min, washed and stored in PBS at 4 °C in dark before imaging. Prior imaging, the coverslips were mounted on imaging chamber and PBS was added to the chamber. Fluorescence lifetime imaging was performed using Leica STELLARIS FALCON confocal microscope equipped with Plan Apochromat ×40/1.25 motCORR glycerol immersion objective. Cells were excited with White Light Laser Stellaris 8 at 450 nm, and fluorescence lifetime times were recorded with HyD X detector, in the range 455 to 495 nm to obtain the photon arrival times specific to donor emission. The pixel-by-pixel photon arrival times were fitted for bi-exponential decay components using n-Exponential Reconvolution fitting model of Leica LAS X software (v. 4.4.0) to obtain mean lifetimes from individual cells.

### Osmotic manipulation of nuclei and hydrogel cushion assay
Nuclei were manipulated by altering the osmolarity of the medium. Hyperosmotic conditions were achieved by adding 250 mM sucrose to the medium. The cells were imaged prior to sucrose treatment and 15 minutes after adding sucrose. The nuclear shrinkage and corresponding change in FRET of Lamin-SS sensor was analyzed. The nuclear volume was calculated from the z-stack images of individual cells using ImageJ Fiji distribution. Briefly, the stack is thresholded to retain only the nuclear region and a short macro was then used to loop through each slice in the stack and measure the area from the thresholded pixels. Finally, the sum of the area from the slices is multiplied with the depth of each slice to obtain the nuclear volume.

Hydrogel cushions were constructed from polyacrylamide (mixing ratio 12.5% acrylamide (#1610140, Bio-Rad Laboratories, Hercules, USA), 7.5% Bis-acrylamide (#1610142, Bio-Rad Laboratories, Hercules, USA). The polymerization of the gel was initiated by adding TEMED (final concentration 0.2% (vol/vol), (#1610800, Bio-Rad Laboratories, Hercules, USA)) and APS (final concentration 1% (vol/vol) (10% (weight/volume) stock solution in PBS, #A3678-100G, Merck, Kenilworth, USA)). The gel was casted on top of a bind-saline (3-(Trimethoxysilyl) propyl methacrylate, Sigma Aldrich) treated 13 mm coverslip placed inside the lid of a 5 ml Eppendorf tube. A second coverslip (18 × 18 mm, 2% Hellmanex treated) was placed on top of the gel to obtain a flat gel surface. Prior to the imaging experiment, the collagen coated coverslips having cultured cells was mounted to the imaging chamber. The cells were treated with 10 μg/ml cytochalasin for 30 min to relax the nuclear lamina and the hydrogel cushion was placed manually on top of the cells. Cells were imaged prior to and 15 min after adding the hydrogel cushion. FRET was calculated from the images using riFRET method.

### Lamin nanobody epitope characterization with mScarlet tagged lamin A truncates
For detecting the Lamin nanobody binding region in Lamin A, different truncated lamin A constructs tagged with mScarlet were generated. For this, initially, the Lamin A/C was amplified with primers having XhoI site, NLS and a linker (GGSGGSGGTSGG (with SpeI site)) in the 5′ region and HindIII site at 3′end. This was cloned to XhoI/HindIII site of pmScarlet-i_C1 plasmid (Addgene Plasmid #85044) to generate mScarlet fusion of the amplified LaminA/C. For different truncated constructs, the sequence of truncated regions was amplified with primers having 5′

SpeI site and 3′ stop codon followed by an XhoI site (description of the primers can be found in the supplementary data 1 provided with this article). This amplified truncated sequence was then subcloned to the SpeI/XhoI site of the mScarlet -Lamin A/C plasmid to replace the Lamin A/C resulting in mScarlet-truncated Lamin fusion constructs. For FRET analysis, the truncated constructs were transiently transfected together with plasmid having EGFP-lamin nanobody to LaminA/C KO cells by using the Neon™ electroporation system (1650 V, 20 ms, 1 pulse; Thermo Fisher Scientific). The cells were fixed after 48 hours with 4% PFA for 10 mins, washed and stored in PBS at 4 °C before imaging. Imaging was done by using a Zeiss LSM 780 laser scanning confocal microscope with a Plan Apochromat ×63/1.4 oil immersion objective, and images were acquired with 488-nm and 561-nm excitation lasers for EGFP and mScarlet, respectively. FRET between EGFP and mScarlet was analyzed using acceptor photobleaching method[72], with 561 nm laser and the images were analyzed with ImageJ Fiji -distribution.

### Statistical analysis
Statistical significance was measured using an unpaired or paired two-tailed Student´s t-test for data containing two groups. For data involving more than two groups, the Ordinary One-way Analysis of Variance (ANOVA) test was performed to obtain the statistical analysis for the data sets concerned. A further comparison of the groups was conducted using the Tukey (HSD) test to obtain significant differences between multiple groups. All statistical tests were conducted at a 5% significance level. GraphPad Prism was used for statistical analyses.

### Reporting summary
Further information on research design is available in the Nature Portfolio Reporting Summary linked to this article.

## Data availability
The microscopy and western blot data generated in this study have been deposited in the IDA database [https://doi.org/10.23729/b1111fbb-5289-4258-a276-42ce002fdb0d] under CC BY 4.0 license. The quantified data used in the figures are provided in the Supplementary Information/Source Data file. Further details of the data can be inquired from the corresponding authors. The Lamin-TS model was constructed from PDB structures 2HQK (mTFP1), 1MYW (Venus), and 4JVP (structure of anti-Hepatitis C virus nanobody). Source data are provided with the manuscript. The description of the primers can be found in the supplementary data 1 provided with this article. Source data are provided with this paper.

## Code availability
SensorFRET analysis tool is available at https://github.com/crmayerVCU/pySensor.

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

## Acknowledgements

We thank Jan Lammerding, Alice Varlet, Andrew Stephens, and Brenton Hoffman for thoughtful discussions and Heidi Peussa for help with the LMNA KO cells. We are grateful to Vesa Hytönen and Vasyl Mykuliak for their help in the sensor structure simulations. The authors acknowledge the Biocenter Finland (BF), Tampere Imaging Facility (TIF), and the VCU Nanomaterials Characterization Core (NCC) for microscopy services. In addition, we wish to acknowledge Tampere University Virus Facility and Eric Dufour for the help in sgRNA design, Flow Cytometry Facility and Laura Kummola for the services and Light Microscopy Unit supported by HiLIFE and BE, Institute of Biotechnology, University of Helsinki, for the FLIM imaging. We are grateful to Jennifer Lippincott-Schwartz lab and Aki Manninen lab for the MDCK cells used in this study. This project was funded in part by a National Science Foundation Graduate Research Fellowship (to B.E.D.), National Science Foundation awards CMMI 1653299 and CMMI 2135653 (to D.E.C.), National Institute of Health award R35 GM119617 (to D.E.C.), as well as Academy of Finland under the award numbers 308315, 314106, 335520 (to T.O.I.), 323507 (to B.G.A.) and 332615 (to E.M).

## Author contributions

Sensor idea was conceived by T.O.I., design and construction of the sensor was conducted by D.E.C. Experimental approach and concepts were designed by T.O.I and D.E.C. Sensor FRET data was generated by B.E.D. and C.R.M. developed the sensor FRET quantification approach. riFRET data was generated and analyzed by B.G.A. Lamina organization studies were conducted by E.M. Sensor expressing cell lines were established by J.I.C. LMNA KO cell line was established by A.R. and E.M., B.G.A. participated in screening. FLIM experiments were conducted by B.G.A. and FRAP experiments by T.O.I., B.G.A. and F.E. Nanobody epitope mapping was conducted by B.G.A. and E.M. F.E. constructed the structural images of the sensor. T.O.I. and D.E.C. wrote the first draft of the manuscript, T.O.I.; D.E.C.; B.E.D.; B.G.A., and E.M. edited the text and generated the finalized version of the manuscript and the supplement.

## Competing interests

The authors declare no competing interests.
