## [Peer Review File · Nature Communications]

REVIEWER COMMENTS

Reviewer #1 (Remarks to the Author):

In this manuscript Danielsson et al. described a novel reagent that allows to study the force that is exerted on protein by using FRET analysis. This reagent should have not influence on the protein levels it is targeting. The applicants apply this reagent to study the nuclear lamins and their interaction with chromatin.

While this approach is very appealing, it seems that additional characterization of this elegant system is needed.

1. The composition of the reagent is not well described. The molecular details should be available. How does it
2. The precise recognition of the nanobodies should be revealed.
3. It is not clear what precisely is measured in in the lamin experiments. It can be inter- or intra- lamin filaments alterations.
4. The assumption that the reagent does not alters lamin expression is reasonable but should be confirmed.
5. The LMNA knockout verification by western blot shows a prominent lane at ~60-65kDa. Knockout should be authenticated by sequencing and not immunofluorescence.
6. The plasmids encode for the reagents used here, should become available for academic use.

Reviewer #2 (Remarks to the Author):

Networks of Intermediate Filament (IF) proteins are generally viewed as conferring important mechanical properties to a cell, and yet there are relatively few molecular-specific measurements of strain sustained by IF networks under stress. The key FRET construct here is a bivalent binder to laminA, which is a key IF in the nucleus; additional constructs include a crosslinker between laminA and a Histone plus control. The approach and observations are interesting. Nonetheless, several initial issues temper enthusiasm:

1. At least one early study used patterned photobleaching of GFP-laminA coupled to imaging as the nucleus was stretched into a pipette (Pajerowski 2007). The images show laminA-specific strains of 10-20% or more, which is sustained. What are the strains sustained here by the FRET approach for the various perturbations? Estimates of max strain should be added here as a bargraph or as a Table for perturbations by osmolarity, ROCK inhibitor, DN-KASH, etc. The authors should indicate where the max strain is observed (envelope, pole, interior, etc) and relate to past studies.

2.

Given the claims here of "measuring the mechanical strain of lamin filaments", more direct mechanical approaches should be used than the approach with ROCK inhibitor or DN-KASH, or osmol changes that might cause complex ion changes. For example, the authors should compress nuclei in cells between two coverslips, which will cause the nuclear rim to increase so that the ratio of nuclear circumference (after/before) provides an easy to calculate strain to relate to FRET signal. Removing the compressing coverslip can also provide important information on reversibility of nuclear shape and FRET.

3. The authors write "large forces were also present on nucleoplasmic lamins", but this requires some clarification of time-scales. FLIM for the biosensor is measured to be nano-seconds as expected, and FRAP timescales for dissociation of the biosensor from the lamina are estimated to be 1~30 sec, which is typical of proteins. However, if lamin epitopes (in the nucleoplasm or even at the lamina) diffuse and displace nanometers on timescales that are shorter, similar, or longer than the dissociation or association time, then the FRET signal should be affected. Moreover, if two distinct lamin filaments are diffusing through chromatin, and then the biosensor binds and crosslinks the two, wouldn't the crosslinking tend to entangle the two lamin filaments in the chromatin, immobilize the complex, and suggest a rigidity that is an artifact of the approach?

4. Fig.4f shows FRET for bivalent and monovalent Histone-LaminA, and the bivalent construct in cells with KO of LaminA should have similar FRET as the monovalent construct. The authors need to indicate the p-value and explain any differences between the sets of KO results.

Reviewer #3 (Remarks to the Author):

Summary:

In this manuscript, Danielsson et al. demonstrate fabrication of nanobody-based FRET biosensors for measuring the tension in the perinuclear lamina as well as nucleoplasmic lamin A/C and chromatin. The authors then use this technology to show that nuclear lamina is under constant force and that these forces rely on the actomyosin contractile state, LINC complex intactness, chromatin condensation state, as well as the nuclear volume and cell cycle. They further demonstrate that likely interactions between nucleoplasmic lamin A/C and chromatin is accompanied by generation of forces between these two systems, which are significantly negated upon KO of the lamin A/C.

I commend the authors for these efforts and find the development of such biosensors highly beneficial to the mechanobiology community. Nonetheless, limitations in understanding how these sensors interact with Lamin A/C meshwork complicates how the results from the study should/may be interpreted. Reasonable efforts to address this issue can reveal the potency and proper applications of these biosensors and are thus, warranted.

Major comments:

- Given that the epitope of the lamin A/C antibody is unknown, a major concern is that whether and to what extent changes in the FRET signal could be affected by real changes in the tension as opposed to the epitope accessibility. As authors mention, this could be further confounded by the fact that lamin A/C epitope accessibility can vary with mechanical forces. For instance, if increased tension in the lamina could somehow block the interacting epitope, how much of the decrease in FRET is due to this accessibility as opposed to the increased tension per se? I understand this is not a trivial task and the authors try to partially address this in Fig.3, but could they think of experiments to get closer to identifying the binding site and interaction mechanism?

- As authors mention in the discussion, it is likely that the sensor nanobodies bind to two lamin A/C proteins in one filament or two lamin A/C sites on adjacent filaments. In the first scenario, the FRET results could be interpreted as indicator of the tension whereas in the latter case, while one or even both filaments maybe under tension and hence lower FRET states, the two filaments maybe getting closer to each other due to overall tension and thus higher FRET states. Identifying the binding site/s accompanied by computational molecular dynamics simulations can clarify what is exactly being measured in the system. Authors should consider these possibilities or at least discuss them in more depth in the Discussion section.

- The reported results for chromatin decondensation using TSA suggests a decrease in lamina tension. Chen et al. (J Cell Biol (2019) 218 (12): 4063–4078) report increase in nuclear size upon chromatin decompaction. Do authors see similar effects in their system? If so, shouldn't this increase in nuclear volume likely elevate the tension in the nuclear lamina and thus lower the FRET state?

Minor comments:

- There is no discussion of Fig. 1e in the text and should be addressed.

Reviewer #4 (Remarks to the Author):

In this manuscript Danielsson et al. describe the development of a new class of protein-based FRET sensors to measure intracellular forces. While traditional sensors, such as vinculin TSmoc or ssFRET, rely on the insertion of the sensor within the protein of interest, the authors used nanobodies fused to the tension-sensitive module to retrieve the forces applied on laminin, more specifically, forces between lamin molecules.

This is, to the best of my knowledge, the first nanobody-based FRET force sensor reported. The idea to measure the force between molecules (intermolecular) is not new, but this is typically done extracellularly, using 'exogenous' sensors (both DNA and protein-based). The idea to have the sensor expressed in living cells but not fused to the protein of interest is very interesting and opens the door to measure intracellular forces on a multitude of targets. Importantly, in contrary to intramolecular sensors, this type of sensor will allow to evaluate forces between different molecules within the same cellular structure (as the example shown in this manuscript, between lamin and histones).

I think that the article is of interest to the broad readership of Nature Communications, but some aspects require some clarification/discussion before publication.

- Crosslinking effects?

By using sensors that can bind to two different lamin molecules, the authors can induce crosslinking between the lamin molecules, which can affect the results obtained (and cause a difference in the mechanical response of the nuclear lamin). Do the authors have any indication if the crosslinking between the molecules does not affect the mechanical response? For instance, is the reduction in the nuclear volume upon osmotic shock similar between cells expressing the SS and TM sensors?

The issue of crosslinking is even more relevant when linking two different molecules. The authors developed a sensor to measure the force between lamin and histone, in fact crosslinking the two structures. The authors should be careful when analyzing the results obtained using this sensor. For instance, on line 212 the authors wrote: "Lamin-histone-SS established that mechanical forces can be transduced between chromatin and A-type lamin". Would the mechanical forces still be transmitted if the sensor was not linking the two structures?

- Force or distance?

FRET measures a distance between two fluorophores. When a FRET-force module is inserted within a protein (intramolecular sensor), there is no reason for the spring to be stretched, so any deviations from the control sensors is due only to force applied across the molecule. However, in the case of intermolecular sensors, the distance between acceptor and donor might also be affected by the distance between the two binding sites. In other words, the sensor will be stretched not because there is a force applied, but because the binding sites lie at a distance only reachable if the sensor is stretched. I think that the FRET efficiency of 14-17 % calculated without any treatment reflects the average distance between the binding sites and not a 'force'. Since the authors detect changes upon different treatments, the sensor is reporting change in the nuclear lamina. However, I think this might be related to changes in the organization/structure of the lamin network rather than changes in the force between lamin molecules. (Of course, the structural changes are a consequence of changes in the mechanical stress). Could the authors comment on this? For the calculated length of the sensor (40nm, mentioned in line 258), what is the expected FRET efficiency? For a FRET efficiency of 27%, what is the distance between the binding sites?

- Terminology regarding FRET calculations

The authors use different methods to calculate FRET efficiency, and it is often not clear which method was used to calculate what... Sometimes there is 'FRET efficiency', sometimes 'apparent FRET eff'. On the methods section, three different methods are used to calculate FRET (sensorFRET, RiFRET, FLIM-FRET) and on the main text other terms are also used, namely spectral-based FRET, FRET ratio, quantified FRET values. The authors should clarify which method was used (even if just in the figure caption) and use uniform definitions.

Are the terms 'FRET eff' and 'Apparent FRET eff' used to refer to different methods to calculate the FRET efficiency?

- Different FRET efficiencies

There are only 4 places in the manuscript where 'Apparent FRET efficiency' is used (Fig. 2c, 4g, S1c, S4). Except for the data shown in Figure 4g, in all the other graphs where 'Apparent FRET eff' is plotted, only the SS sensor is shown. In figure 4g, the 'Apparent FRET eff' of the lamin-histone TM sensor is 0.261, while in the other experiments, the FRET efficiency of lamin TM sensor was around 0.4. This seems to suggest that the two different ways to calculate FRET efficiency give a different value. On top of this, if we calculate the FRET efficiency from the change in the lifetime [using the data in Fig. 1d and the equation $\text{FRETeff} = 1 - (\text{lifetime DA})/(\text{lifetime D})$], we get values of 8 and 24 %, from SS and TM (different than the ones shown throughout the manuscript). And, on Figure 4b, the lifetime of the TM sensor is different than the lifetime reported in figure 1d (2.35 versus 2.10) – which would result in a different FRET efficiency...

It is clear to me that it is difficult to give an absolute value for FRET efficiency, but in that case, I would suggest calculating FRET efficiencies always with the same method (sensorFRET or RiFRET). If it is not possible to do this, the differences obtained using different methods should be discussed (at least a comment acknowledging that these differences arise from the different analysis used).

- FRAP measurements

The authors used FRAP measurements to evaluate the binding kinetics of both SS and TM sensors. They used a model that assumes that (1) the binding of both nanobodies is the same and (2) the sensor binds to an immobile molecule. In the methods section (sensor design), the authors say that the “the C-terminal lamin A nanobody was designed by using the reverse sequence of the VHH” – do the authors know if this has the same binding affinity? If an experiment is made using the sensor with only the C-terminal nanobody, will the FRAP data be similar? Regarding point (2), how will be the FRAP data of fluorescently labelled lamin molecules?

To evaluate the diffusion of the sensors in the nucleus, it would also be interesting to perform FRAP measurements using the KO cell line (where no binding is expected) or TSmcd without any nanobody, but still targeted to the nucleus with a NLS sequence.

The reason why I bring up this discussion is because the fitting of the curve of the TM sensor is not very accurate (supporting figure 2). There seems to be a second fraction that is not taken along in the fitting. Since the authors discuss the binding dynamics extensively in the final section, it would be important to make sure that the model used to fit the FRAP data is accurate.

- Data comparison

FRET efficiency for both sensors are measured in a lot of different conditions. It would be interesting to see a Table summarizing of the values obtained (instead of mentioning them in the text, at least for the controls). This would also allow to discuss which of the outside/internal stimuli has a stronger impact in the nuclear lamina. I missed the discussion of what treatment leads to a more pronounced effect.

Minor comments:

- What is the force threshold for the interaction between the nanobodies and lamin? Would this explain the minimum FRET efficiency calculated? (Would the sensor detach before it stretches to its maximum?)
- Was there any special reason why the authors used MDCK cells? If yes, please add.
- According to the methods section the authors used Airyscan microscopy to obtain images with a higher resolution. Can you indicate on the figure caption which images were acquired with this microscope?
- Is the panel b in Figure 1 a xy-projection of a z-stack? If yes, can you add this information on the figure caption?
- For the images evaluating overlap (Figs. 1b, 3f, 3g, 4d), would it be possible to perform so image correlation in addition to line profiles? For figures 3f and 3g, it is not clear what is plotted in the graphs (c-term / rod-domain).

- Figure 1e reports intensity but uses the same colormaps as the one used for FRET. For clarity, I would change this to grey or cyan colormap (as used for the other 'intensity-based' images).
- Why is the distribution of the apparent FRET efficiency in supporting figure S1 broader than in Figure 1c? They are both for lamin-TS expressed in MDCK WT, no? Is this linked to the analysis method?
- I miss information on how the nuclear volume was calculated – can you add this information on the methods section?
- For the experiment with the changes on the osmotic pressure – what is the effect of this treatment on the TM sensor?
- Is the data shown in figure 1g coming from the same cell? If so, there is a small difference in the apparent FRET efficiency but a big difference in the morphology of the nucleus. I would expect that these morphological changes would be reflected in the FRET value measured – can the authors comment on this?
- On the beginning of the section regarding cell cycle, EMT and chromatin condensation, the authors wrote “Detecting a large heterogeneity in the quantified FRET values of Lamin-SS between individual cells (line 151)” – which data does this sentence refer to?
- What was the rationale to evaluate the effect of EMT on nuclear lamin? This was not clear from the text (section starting on line 160).
- On line 179 the authors refer to “ratiometric imaging” to quantify the antibody labelling ratio. It was not clear how this analysis was done. In addition, I would try to analyze the images using image correlation methods (eg. Pearson correlation) – see previous comment.
- Line 202: “This sensor [Lamin-histone-SS] exhibited a more pronounced predominant nucleoplasmic localization” – what do the authors mean with this?
- Line 205: “FRET efficiency of 17% (mean 0.27...)” is this correct? Looking at the figure (4e) and mean value, I would say it is 27%...
- Line 226: I would not say that the authors “demonstrate that the lamin A/C network experiences significant force, which is regulated by actin, myosin, and the nuclear-cytoskeletal connections”. I think the authors show that is ‘force’ is affected by these parameters, but not that it is regulated (small detail). [‘force’ or distance...]
- Line 248: “our histone-lamin sensor shows that nucleoplasmic lamins, as opposed to lamins at the periphery, are the primary lamin component mechanically interacting with chromatin”. I think the interaction with the chromatin is caused by the presence of the nanobody against histone in the sensor (the sensor will only be present where there is chromatin), and therefore the authors cannot say that the lamin at the periphery does not interact with chromatin (there might be a physical interaction).
- Line 287: “it is possible that mechanical forces applied across the sensor causes one or more of the fluorescent proteins to unfold”. I find this unlikely, especially given the reversible nature of the binding of the nanobody. I would expect that the sensor would unbind rather than unfold (it would be important to have the values of the forces required for unbinding and unfolding to support this hypothesis). Also – what would unfold first: the nanobody or the fluorescent protein?

- On the same sentence, the authors say that unfolding of the proteins would result in decreased fluorescence of the protein (agreed) and therefore reduced FRET - not sure if I agree 100% with this. I think this depends on how the FRET is calculated (lifetime-based FRET analysis would not be affected) and which of the proteins, donor or acceptor, unfolds.
- At the end of the discussion, the authors say that these strategies could be use with nanobody targeted to GFP and RFP proteins. But, in that case, what would be the advantage compared to intramolecular sensors? I thought that the biggest advantage of the sensors developed here was to avoid adding fluorescent proteins to specific proteins, as this could affect their functionality (line 58/59)
- For the data analysis after the treatments (with CytoD, Y-27632, sucrose...), what was the time between addition of treatment and data acquisition? 20 minutes in all the cases?
- To evaluate if chromatin condensation affect the lamin A/C organization, the authors used two different antibodies that recognized two different epitopes. Strangely, on figure 4f, there seems to be a link between the distribution of the antibody against the rod domain and the expression of the lamin-SS sensor: there are 3 cells in the ROI that are not expressing the SS sensor; in this cells, the anti-rod antibody seems to be more localized on the periphery of the nucleus. Is this a coincidence? Could the authors show/analyse the distribution of the anti-rod AB (in comparison with the C-term) in WT cells and cells expressing the lamin-SS sensor? This might give an indication of where the nanobody binds...
- On figure 4g, there seems to be a statistically significant difference between the SS sensor measured in KO cells and the TM sensor (in both WT and KO cells). Could the authors explain where this difference might come from?

On a small note, I would like to thank the authors for immediately sharing these sensors with the scientific community by depositing them in Addgene.

Rebuttal letter

(our responses are in *italics*)

Reviewer #1 (Remarks to the Author):

In this manuscript Danielsson et al. described a novel reagent that allows to study the force that is exerted on protein by using FRET analysis. This reagent should have not influence on the protein levels it is targeting. The applicants apply this reagent to study the nuclear lamins and their interaction with chromatin.

While this approach is very appealing, it seems that additional characterization of this elegant system is needed.

1. The composition of the reagent is not well described. The molecular details should be available. How does it

The reviewer is correct. We added a more detailed description of the sensor to the beginning of the results section and added a supplemental figure showing the molecular structure of the construct and size of the sensor (lines 95-99 and 113-116 and supplemental figure S1).

2. The precise recognition of the nanobodies should be revealed.

The reviewer raises an important point, not just regarding the manuscript in question but the overall community using the lamin nanobody based tagging of nuclear lamina. We first approached Chromotek-company and asked for the details regarding the epitope. We learned from them that the epitope is unknown, and they have tried to unsuccessfully map it in vitro, by using purified lamin proteins. Based on this we established an assay, where we used a pulldown approach of EGFP-tagged lamin nanobody from MDCK cells and subsequent chemical crosslinking. We hoped to be able to pull down the lamin nanobody together with lamin protein, chemically cross link them together and then distinguish the region where nanobody binds after a mass-spectrometry analysis of the crosslinked proteins. Unfortunately, this assay was also unsuccessful, and we did not detect any nanobody - lamin crosslinks (only lamin-lamin crosslinks). Finally, we approached the binding site problem by using lamin nanobody expressing cells together with various lamin A/C antibodies (competitive binding assay) and truncated lamin A/C proteins (FRET binding assay). These experiments indicated that the nanobody binds to a region near the end of the coiled coil domain. We have added this information to the beginning of the result section and to a supplemental figure (lines 99 - 109, supplemental figure S2).

3. It is not clear what precisely is measured in in the lamin experiments. It can be inter- or intra- lamin filaments alterations.

This is an excellent question. We have now modeled the sensor size and used the high-resolution electron microscopy images of nuclear lamina by Turgay et al. (<https://doi.org/10.1038/nature21382>) to map the possible binding radius of the sensor. The nanobody binds to a single binding site in the target protein and is essentially a monoclonal antibody. Furthermore, lamin filaments have repetitive structure with approximately 20 nm intervals. Based on this information, the model indicates that the sensor is able to bind to single filaments or bridge two separate lamin A/C filaments. Thus, the sensor can measure strain in single filaments or more widely within the nuclear lamina. We have added discussion related to this (lines 113-126 and supplemental figure S1 and S3).

4. The assumption that the reagent does not alters lamin expression is reasonable but should be confirmed.

We performed western blotting of lamin A/C from wildtype MDCK cells and from MDCK cells stably expressing Lamin-SS sensor. The data indicates that the lamin A/C expression is not changed. We have added data to the supplement and mention this also in the results section (lines 133-134, supplemental figure S1).

5. The LMNA knockout verification by western blot shows a prominent lane at ~60-65kDa. Knockout should be authenticated by sequencing and not immunofluorescence.

Thanks for pointing this out. When looking at the data carefully, it seems that the lanes which are visible in the KO cells are also visible in the wildtype cells and they don't overlap with the signal coming from lamin A/C. However, we totally agree that the KO verification needs to be more thorough. Therefore, we also performed sequencing, as proposed by the reviewer, and saw a frameshift mutation in the lamin A/C gene. We also performed immunolabeling by using an antibody which recognizes the N-terminal epitope of lamin A/C and saw negative staining in KO cells when compared to wt cells. We have added data to the supplement and mention this also in the results section (lines 143-151, supplemental figure S4).

6. The plasmids encode for the reagents used here, should become available for academic use.

We agree with the reviewer. Unfortunately, our actual plan to distribute the sensors via Addgene had to be changed. Company Proteintech (previously Chromotek) contacted us regarding the sensor (after publication of bioRxiv preprint) and stated that they own the intellectual property rights of the used nanobodies and argued that based on the end user license agreement (EULA) we can't distribute the sensor freely. However, Proteintech understood well our needs and we have now reached a tentative agreement

where Proteintech will distribute the sensors with non-profit basis (charging a small fee to recover distribution costs). This ensures that the sensors will be easily accessible to the scientific community and allowing Proteintech to preserve their intellectual property rights. Thus, similarly to the paper Virant et al. (<https://doi.org/10.1038/s41467-018-03191-2>) we added the following text to the manuscript (lines 450-452):

“The nanobody was previously developed by Rothbauer et al²⁴ and the expression vector coding for the lamin-nanobody was provided by ChromoTek & Proteintech (Planegg-Martinsried, Germany) with a corresponding end user license agreement.”

Reviewer #2 (Remarks to the Author):

Networks of Intermediate Filament (IF) proteins are generally viewed as conferring important mechanical properties to a cell, and yet there are relatively few molecular-specific measurements of strain sustained by IF networks under stress. The key FRET construct here is a bivalent binder to laminA, which is a key IF in the nucleus; additional constructs include a crosslinker between laminA and a Histone plus control. The approach and observations are interesting. Nonetheless, several initial issues temper enthusiasm:

1. At least one early study used patterned photobleaching of GFP-laminA coupled to imaging as the nucleus was stretched into a pipette (Pajeroski 2007). The images show laminA-specific strains of 10-20% or more, which is sustained. What are the strains sustained here by the FRET approach for the various perturbations?

This is an excellent question. The strain of the nuclear envelope, and possible correlation of the strain and measured FRET value would be highly interesting. However, the imaging requirements regarding FRET limit the other approaches which can be combined to the experiments. Dyes which work in the far-red region of the visible spectrum (excitable e.g., with 633 nm laser) can in principle be used together with the sensor. Unfortunately, different photoactivation, -conversion or -photobleaching experiments are highly challenging since they would also affect either the donor or acceptor of TSmod. Thus, the more detailed studies of local strains in the nuclear lamina are currently difficult. However, based on the quantification of nucleus morphology (please see below) we expect the changes in overall nuclear strain or morphology to be fairly small when using pharmaceutical drugs.

Estimates of max strain should be added here as a bargraph or as a Table for perturbations by osmolarity, ROCK inhibitor, DN-KASH, etc. The authors should indicate where the max strain is observed (envelope, pole, interior, etc) and relate to past studies.

This is again a good point. Quantification of the strain is challenging together with the FRET measurements, since additional fluorescent markers are difficult to combine and excessive imaging leads into bleaching of the sensor, altering the FRET. However, we quantified the changes in nucleus morphology during Y-27632 and TSA inhibition, and with dominant negative KASH (DN-KASH) expression. ROCK inhibition did not affect the

nucleus circularity but it led to reduced nucleus cross-sectional area, indicating possible relaxation of the nuclear lamina strain. Histone deacetylase inhibition or DN-KASH expression did not alter nucleus circularity or cross-sectional area, suggesting that the changes in the strain are more subtle. We have added supplemental data regarding the nucleus morphology and edited the results section accordingly (results section lines 207-211, 223-2226 and 261-262, Supplemental figures 8 and 9).

In addition, currently the signal-to-noise ratio of the sensor does not allow high resolution spatial mapping of the FRET. We believe that more detailed and quantitative studies of nucleus (local) strain and subsequent local FRET are beyond this study.

2. Given the claims here of "measuring the mechanical strain of lamin filaments", more direct mechanical approaches should be used than the approach with ROCK inhibitor or DN-KASH, or osmol changes that might cause complex ion changes. For example, the authors should compress nuclei in cells between two coverslips, which will cause the nuclear rim to increase so that the ratio of nuclear circumference (after/before) provides an easy to calculate strain to relate to FRET signal. Removing the compressing coverslip can also provide important information on reversibility of nuclear shape and FRET.

Thanks for the suggestion. We performed direct mechanical manipulation of the nuclei by compressing cells with polyacrylamide hydrogel, similarly to Ihalainen, Aires et al. (<https://doi.org/10.1038/nmat4389>). This allowed us to observe the opposite effect to osmotic shock. The osmotic shock led to shrinking of the nuclei and relaxation of the nuclear lamina (increased FRET of the Lamin-SS sensor), but gel compression caused nucleus to stretch and tensed lamina (reduced FRET of the Lamin-SS sensor). We have added these results to the manuscript and data to the figures (lines 185-197, figure 2).

3. The authors write "large forces were also present on nucleoplasmic lamins", but this requires some clarification of time-scales. FLIM for the biosensor is measured to be nano-seconds as expected, and FRAP timescales for dissociation of the biosensor from the lamina are estimated to be 1~30 sec, which is typical of proteins. However, if lamin epitopes (in the nucleoplasm or even at the lamina) diffuse and displace nanometers on timescales that are shorter, similar, or longer than the dissociation or association time, then the FRET signal should be affected. Moreover, if two distinct lamin filaments are diffusing through chromatin, and then the biosensor binds and crosslinks the two, wouldn't the crosslinking tend to entangle the two lamin filaments in the chromatin, immobilize the complex, and suggest a rigidity that is an artifact of the approach?

The origin of the strain in the sensor is an interesting question. Lamins in the nuclear lamina are probably almost immobile during the binding time of the sensor. In addition, random diffusion should increase the epitope-to-epitope distance in some cases and reduce in others, thus the overall effect on the measured FRET is difficult to predict. In the nucleoplasm the chromatin diffusion coefficient is about $0.1 \mu\text{m}^2/\text{s}$. Thus, during the dual binding time of the sensor the chromatin can diffuse a few tens of nanometers. However, the effect of diffusion on the sensor or the force rising from diffusive

movement of the epitopes is hard to estimate. This is still a valid point and we have added discussion related to this (discussion lines 406-408).

4. Fig.4f shows FRET for bivalent and monovalent Histone-Lamina, and the bivalent construct in cells with KO of LaminA should have similar FRET as the monovalent construct. The authors need to indicate the p-value and explain any differences between the sets of KO results.

Correct observation. We also noticed that the Lamin-SS should have a higher FRET in LMNA KO cells. We dont have any direct explanation for this, but we believe that the difference rises from the constructs themselves, different cell lines, and sensor transfections. Lleres et al. were able to use FRET between histone fluorescent proteins to measure the compaction state of the chromatin (<https://doi.org/10.1083%2Fjcb.200907029>). Thus, the monovalent Histone-Lamina sensor might lead into higher intermolecular FRET between only histone binding sensors. In addition, the two nanobodies in the full-length version of the sensor might also limit the free movement of the fluorescent proteins and in monovalent sensor the proteins (donor and acceptor) are more mobile, leading to higher FRET. We have added a short discussion of this in the end of the results section (lines 310-314).

Overall, the data indicates the importance of comparisons between the SS and TM sensors when doing the FRET measurements. However, we want to highlight that the changes in FRET were not detected with TM construct.

Reviewer #3 (Remarks to the Author):

Summary:

In this manuscript, Danielsson et al. demonstrate fabrication of nanobody-based FRET biosensors for measuring the tension in the perinuclear lamina as well as nucleoplasmic lamin A/C and chromatin. The authors then use this technology to show that nuclear lamina is under constant force and that these forces rely on the actomyosin contractile state, LINC complex intactness, chromatin condensation state, as well as the nuclear volume and cell cycle. They further demonstrate that likely interactions between nucleoplasmic lamin A/C and chromatin is accompanied by generation of forces between these two systems, which are significantly negated upon KO of the lamin A/C.

I commend the authors for these efforts and find the development of such biosensors highly beneficial to the mechanobiology community. Nonetheless, limitations in understanding how these sensors interact with Lamin A/C meshwork complicates how the results from the study should/may be interpreted. Reasonable efforts to address this issue can reveal the potency and proper applications of these biosensors and are thus, warranted.

Major comments:

- Given that the epitope of the lamin A/C antibody is unknown, a major concern is that whether and to what extent changes in the FRET signal could be affected by real changes in the tension as opposed to the epitope accessibility. As authors mention, this could be further

confounded by the fact that lamin A/C epitope accessibility can vary with mechanical forces. For instance, if increased tension in the lamina could somehow block the interacting epitope, how much of the decrease in FRET is due to this accessibility as opposed to the increased tension per se? I understand this is not a trivial task and the authors try to partially address this in Fig.3, but could they think of experiments to get closer to identifying the binding site and interaction mechanism?

This is an important topic related to the functionality of the sensor, since changes in the epitope accessibility can theoretically affect the resulting FRET. We have now characterized the nanobody epitope and according to our antibody competition assay and FRET experiments the nanobody binds to the end of the coiled coil domain of lamin A/C. The accessibility of this region of the lamin A/C has not been previously shown to be force dependent and the force sensitive regions are situated more to the N- and C-terminus of the protein (e.g., 131C3 and JOL-2 antibody epitopes). In addition, we did not see correlation between Lamin-SS and force sensitive lamin A/C antibody 131C3 or apicobasal polarization of the Lamin-SS labeling, similar to 131C3 labeling in Ihalainen, Aires et al. (<https://doi.org/10.1038/nmat4389>). Thus, we can't directly exclude the possibility of the changes in the nanobody epitope accessibility, but our data does not indicate that it would be a major factor influencing the measured FRET.

We have added information regarding the sensor functionality to the beginning of the result section and to a supplemental figure (lines 98-109 and 130-133, supplemental figure S1-S3).

- As authors mention in the discussion, it is likely that the sensor nanobodies bind to two lamin A/C proteins in one filament or two lamin A/C sites on adjacent filaments. In the first scenario, the FRET results could be interpreted as indicator of the tension whereas in the latter case, while one or even both filaments maybe under tension and hence lower FRET states, the two filaments maybe getting closer to each other due to overall tension and thus higher FRET states. Identifying the binding site/s accompanied by computational molecular dynamics simulations can clarify what is exactly being measured in the system. Authors should consider these possibilities or at least discuss them in more depth in the Discussion section.

Reviewer is correct that it's important to understand how the sensor works at the molecular level. We have initiated a separate project where we investigate the sensor behavior by using steered molecular dynamics (SMD) simulations. Unfortunately, the large sensor size (over 850 aa) makes the SMD simulation work challenging, and we have focused mainly to the TSmod unfolding and refolding. Thus, we feel that the modelling of the binding of the whole sensor to nuclear lamina is beyond this study. However, in addition to the epitope characterization (see above), we also modeled the sensor binding to nuclear lamina and according to the model the sensor can bind to single filaments or bridge adjacent filaments. We have added discussion related to this topic (lines 113-126, supplemental figures S1 - S3).

- The reported results for chromatin decondensation using TSA suggests a decrease in lamina tension. Chen et al. (J Cell Biol (2019) 218 (12): 4063–4078) report increase in nuclear size upon chromatin decompaction. Do authors see similar effects in their system? If so, shouldn't this increase in nuclear volume likely elevate the tension in the nuclear lamina and thus lower the FRET state?

This is an important point. The effect of chromatin compaction and TSA on nuclear size is a complex issue. Nucleus size has been reported to increase when chromatin is decondensed (Bustin, M., and T. Misteli <https://doi.org/10.1126/science.aad6933>). However, TSA treatment can also reduce the nucleus volume (Supplementary data in Fischer et al. <https://doi.org/10.3389/fcell.2020.00393>). Often the TSA treatment is conducted for 24 h and this might also alter gene expression, leading to indirect changes in nuclear organization. We performed 4 h TSA treatment to avoid these effects. Motivated by this comment we now performed quantification of nuclear perimeter and cross-sectional area to approximate the TSA induced changes in nuclear mechanics. According to our data, TSA treatment did not have an effect on the nucleus circularity or cross-sectional area (Supplemental figure S9).

Minor comments:

- There is no discussion of Fig. 1e in the text and should be addressed.

Thanks for pointing out this error. We have now added a short discussion of the Fig. 1e to the manuscript (lines 157-160).

Reviewer #4 (Remarks to the Author):

In this manuscript Danielsson et al. describe the development of a new class of protein-based FRET sensors to measure intracellular forces. While traditional sensors, such as vinculin TSmod or ssFRET, rely on the insertion of the sensor within the protein of interest, the authors used nanobodies fused to the tension-sensitive module to retrieve the forces applied on laminin, more specifically, forces between lamin molecules.

This is, to the best of my knowledge, the first nanobody-based FRET force sensor reported. The idea to measure the force between molecules (intermolecular) is not new, but this is typically done extracellularly, using 'exogeneous' sensors (both DNA and protein-based). The idea to have the sensor expressed in living cells but not fused to the protein of interest is very interesting and opens the door to measure intracellular forces on a multitude of targets. Importantly, in contrary to intramolecular sensors, this type of sensor will allow to evaluate forces between different molecules within the same cellular structure (as the example shown in this manuscript, between lamin and histones).

I think that the article is of interest to the broad readership of Nature Communications, but some aspects require some clarification/discussion before publication.

- Crosslinking effects?

By using sensors that can bind to two different lamin molecules, the authors can induce crosslinking between the lamin molecules, which can affect the results obtained (and cause a

difference in the mechanical response of the nuclear lamin). Do the authors have any indication if the crosslinking between the molecules does not affect the mechanical response?

This is a good question. Since the lamin nanobodies bind directly to lamin filaments, they can also mask binding sites of endogenous lamin binding partners and affect cellular physiology. We have not directly compared the cell mechanics between WT and Lamin-SS expressing cells, and therefore we can't exclude this possibility. However, we are not able to detect morphological changes in the nuclei of the Lamin-SS expressing cells. Furthermore, the sensor expression was easy to stabilize in MDCK cells and after that we constructed a fibroblast cell line expressing the sensor. Finally, we have now used the Lamin-SS sensors also in hPSC derived cardiomyocytes, which still after expressing the sensor show beating and nuclear force transduction. These cells and their functionality are really sensitive for even small disturbances. Together these indicate that the effect of Lamin-SS expression has only a small effect on cellular physiology.

For instance, is the reduction in the nuclear volume upon osmotic shock similar between cells expressing the SS and TM sensors?

This is an interesting question, but would require a considerable amount of work. Lamin-SS and Lamin-TM sensor expressing cells are two separate cell lines, originating from single cell clones. Due to experimental variability the accurate quantification of the possible minute differences of nuclear volume regulation would be challenging.

The issue of crosslinking is even more relevant when linking two different molecules. The authors developed a sensor to measure the force between lamin and histone, in fact crosslinking the two structures. The authors should be careful when analyzing the results obtained using this sensor. For instance, on line 212 the authors wrote: "Lamin-histone-SS established that mechanical forces can be transduced between chromatin and A-type lamin". Would the mechanical forces still be transmitted if the sensor was not linking the two structures?

Reviewer is correctly pointing out the risks related to the intermolecular sensor and crosslinking of different proteins. We would not recommend constructing a sensor between proteins which are not known to physically interact. However, A-type lamins are known to bind chromatin via core histones with high affinity (K_d of 100-300nM) (<https://doi.org/10.1083/jcb.131.1.33> and <https://doi.org/10.1242/jcs.03325>). Thus, the lamin-chromatin interaction allows the force transduction between these structures and therefore the sensor does not create uncommon cross-links. We added short description of the lamin-chromatin interaction to the manuscript (lines 293-295)

- Force or distance?

FRET measures a distance between two fluorophores. When a FRET-force module is inserted within a protein (intramolecular sensor), there is no reason for the spring to be stretched, so any deviations from the control sensors is due only to force applied across the molecule.

However, in the case of intermolecular sensors, the distance between acceptor and donor might also be affected by the distance between the two binding sites. In other words, the sensor will be stretched not because there is a force applied, but because the binding sites lie at a distance only reachable if the sensor is stretched.

I think that the FRET efficiency of 14-17 % calculated without any treatment reflects the average distance between the binding sites and not a 'force'. Since the authors detect changes upon different treatments, the sensor is reporting change in the nuclear lamina. However, I think this might be related to changes in the organization/structure of the lamin network rather than changes in the force between lamin molecules. (Of course, the structural changes are a consequence of changes in the mechanical stress). Could the authors comment on this?

Reviewer is correct that the low FRET efficiencies indicate that there is considerable strain in the sensor already in control conditions. We speculate that this is due to the epitope-to-epitope distance, which is higher than the length of the relaxed sensor. However, this does not affect the overall conclusions or the functionality of the sensor. Increased FRET indicates that the epitope-to-epitope distance is shorter and there is a reduction in the strain.

For the calculated length of the sensor (40nm, mentioned in line 258), what is the expected FRET efficiency? For a FRET efficiency of 27%, what is the distance between the binding sites?

The approximated maximum length of the sensor (41 nm) can be reached only with totally extended linkers and TSmod (see supplementary figure S1). Thus, in that condition the FRET would be negligibly small. Due to the linker regions between fluorescent proteins and TSmod, it is difficult to approximate the distance between the binding sites when certain FRET efficiency would be reached. We assume that the linkers are flexible, but we don't have data on their force-extension behavior. Like stated in the comments to reviewer #3, we have started simulation work related to the sensor behavior, but we are focusing on TSmod unfolding and folding kinetics, since the whole sensor is too large for SMD simulations.

- Terminology regarding FRET calculations

The authors use different methods to calculate FRET efficiency, and it is often not clear which method was used to calculate what... Sometimes there is 'FRET efficiency', sometimes 'apparent FRET eff'. On the methods section, three different methods are used to calculate FRET (sensorFRET, riFRET, FLIM-FRET) and on the main text other terms are also used, namely spectral-based FRET, FRET ratio, quantified FRET values. The authors should clarify which method was used (even if just in the figure caption) and use uniform definitions.

Are the terms 'FRET eff' and 'Apparent FRET eff' used to refer to different methods to calculate the FRET efficiency?

Thanks for pointing out this inconsistency. We have used 3 different methods and 3 different confocal microscopes to quantify FRET, spectral imaging-based sensor FRET (<https://doi.org/10.1038/s41598-017-15411-8>), ratiometric imaging-based riFRET

<https://doi.org/10.1002/cyto.a.20747>) and donor lifetime-based FLIM-FRET. We have now renamed the FRET data in the manuscript and in the figures to sFRET (sensor FRET), riFRET (ratiometric FRET) and FLIM-FRET.

- Different FRET efficiencies

There are only 4 places in the manuscript where 'Apparent FRET efficiency' is used (Fig. 2c, 4g, S1c, S4). Except for the data shown in Figure 4g, in all the other graphs where 'Apparent FRET eff' is plotted, only the SS sensor is shown. In figure 4g, the 'Apparent FRET eff' of the lamin-histone TM sensor is 0.261, while in the other experiments, the FRET efficiency of lamin TM sensor was around 0.4. This seems to suggest that the two different ways to calculate FRET efficiency give a different value. On top of this, if we calculate the FRET efficiency from the change in the lifetime [using the data in Fig. 1d and the equation $\text{FRET}_{\text{eff}} = 1 - (\text{lifetime DA})/(\text{lifetime D})$], we get values of 8 and 24 %, from SS and TM (different that the ones shown throughout the manuscript). And, on Figure 4b, the lifetime of the TM sensor is different than the lifetime reported in figure 1d (2.35 versus 2.10) – which would result in a different FRET efficiency...

It is clear to me that it is difficult to give an absolute value for FRET efficiency, but in that case, I would suggest calculating FRET efficiencies always with the same method (sensorFRET or RiFRET). If it is not possible to do this, the differences obtained using different methods should be discussed (at least a comment acknowledging that these differences arise from the different analysis used).

Reviewer is correct about the different FRET values. Quantification of the FRET can be really challenging, and different methods often give different values. We also used 3 different imaging systems, which can also influence the quantified FRET values. Therefore, we aimed to always use the truncated mutant sensor as a control, so that the data sets are comparable to each other. We see the different quantification methodologies also as a strength of the manuscript since we see similar behavior of the sensor with different FRET quantification strategies. We briefly discuss the different approaches in the discussion-section (lines 330-333).

- FRAP measurements

The authors used FRAP measurements to evaluate the binding kinetics of both SS and TM sensors. They used a model that assumes that (1) the binding of both nanobodies is the same and (2) the sensor binds to an immobile molecule. In the methods section (sensor design), the authors say that "the C-terminal lamin A nanobody was designed by using the reverse sequence of the VHH" – do the authors know if this has the same binding affinity? If an experiment is made using the sensor with only the C-terminal nanobody, will the FRAP data be similar? Regarding point (2), how will be the FRAP data of fluorescently labelled lamin molecules?

Regarding point (1), if the binding of both nanobodies is the same: The reviewer is correct that we are assuming that the kinetics of the binding of the single or dual nanobody occurs with similar kinetics, at least with regard to the binding of a single nanobody. We of course could speculate that there may be some steric differences in

the diffusivity given the slightly larger dual nanobody sensor vs the single nanobody sensor (~10 kDa size difference, Lamin-SS is about 95 kDa and Lamin-TM 85 kDa), but the difference to the diffusion would be really small.

*We also note that the reverse sequence of the nanobody was deliberate. Nanobodies exist as four domains: N-FR1-FR2-FR3-FR4-C. To our knowledge, all chromobodies are constructed in such a manner that the fluorescent protein is attached at the C-terminal end of the nanobody, most proximal to the FR4 domain (N-FR1-FR2-FR3-FR4-linker-fluorescent protein-C). In order to preserve this structure for both nanobodies the sequence was reversed for the second nanobody, with an overall sensor design of: N-FR1-FR2-FR3-FR4-linker-TSmod-linker-FR4-FR3-FR2-FR1-C. This allows the FR1 domain of each nanobody to be most distal from TSmod, providing near-identical structures of each nanobody in terms of accessibility to the epitope binding domains. We believe that the reverse sequence (N-TSmod-FR4-FR3-FR2-FR1-C) provides the closest binding affinity to the first nanobody (N-FR1-FR2-FR3-FR4-TSmod-C). We acknowledge that we did not directly compare FRAP results for a single nanobody in the reverse configuration (TSmod-FR4-FR3-FR2-FR1) with TSmod to a single nanobody in the forward configuration (FR1-FR2-FR3-FR4-TSmod). However, even if there were a slight difference given the presence of an amine vs carboxyl group, we would predict that the alternate “non-reversed” sensor design (FR1-FR2-FR3-FR4-TSmod-**FR1-FR2-FR3-FR4**) is more likely to have differences in the binding kinetics between the two nanobodies given the differences in the positioning of the nanobody domains.*

Regarding point (2), if lamin A is an immobile molecule: During the approximately 1-minute time course of the FRAP experiments, A-type lamin mobility was considered to be minimal, since the half recovery time of lamin A has been reported to be 140 minutes (<https://doi.org/10.1186/1471-2121-5-46>). We also now performed FRAP experiments with EGFP-lamin A expressing MDCK cells and the data indicated negligible recovery during the 1-minute recovery time. Thus, when taken together, we assume that the sensor binding target did not move during the experiment. We have clarified the text in the results section and added data to the figures (lines 160-163, Figure 1 and supplemental figure S5).

To evaluate the diffusion of the sensors in the nucleus, it would also be interesting to perform FRAP measurements using the KO cell line (where no binding is expected) or TSmod without any nanobody, but still targeted to the nucleus with a NLS sequence.

This is an interesting suggestion. We performed FRAP experiments of the Lamin-SS in the LMNA KO cells. In this case the fluorescence recovery was diffusion limited, most probably since the nanobody binding partner was absent. This allowed us to define the diffusion coefficient of the free sensor inside the nucleus and show the difference in recovery when binding is not present. We added the results to the manuscript and to the supplements lines 159-160 and 167-169, Figure 1, Supplemental figure S5).

The reason why I bring up this discussion is because the fitting of the curve of the TM sensor is not very accurate (supporting figure 2). There seems to be a second fraction that is not taken along in the fitting. Since the authors discuss the binding dynamics extensively in the

final section, it would be important to make sure that the model used to fit the FRAP data is accurate.

Reviewer highlights an important aspect regarding FRAP experiments. There are at least two factors influencing the curve fitting and the whole FRAP analysis. First, the recovery of the sensor is initially fast, and qualitatively it is clear that a large fraction of the sensor is diffusing freely or binding highly dynamically. Originally, we did not know the diffusion coefficient of the sensor, and it was approximated based on the sensor size and the known EGFP diffusion coefficient in the nucleus. This probably leads to small errors in the early timepoints of the recovery and modeling of the single bound sensor dynamics. The approximated diffusion coefficient of the sensor was now replaced by the diffusion coefficient which was measured (and modeled) in LMNA KO cells (see above). This allows much more accurate modeling of the sensor binding. However, it is obvious from the data that the TM sensor lacks the slowly recovering fraction, which we interpreted to be the dual bound sensor. This slowly recovering population is visible in the SS sensor recovery data, especially in the fluorescence recovery during 5-30s. Secondly, quantification of the FRAP recovery needs to be conducted via simulating the whole process (bleaching, imaging, binding, diffusion, etc.), since all the process parameters are difficult to include in analytical mathematical equations. We used Virtual Cell -software to conduct the simulations. Unfortunately, the simulations are relatively heavy and take 5-10 min each, thus the parameter space of what can be covered is small. Therefore, precise parameter fitting is challenging.

- Data comparison

FRET efficiency for both sensors are measured in a lot of different conditions. It would be interesting to see a Table summarizing of the values obtained (instead of mentioning then in the test, at least for the controls). This would also allow to discuss which of the outside/internal stimuli has a stronger impact in the nuclear lamina. I missed the discussion of what treatment leads to a more pronounced effect.

Thank you for the suggestion. We added a summarizing table and included to comparison to results section (lines 282-285, Figure 4). However, one needs to be careful when interpreting the differences, since they might also rise from the efficacy of the different drugs.

Minor comments:

- What is the force threshold for the interaction between the nanobodies and lamin? Would this explain the minimum FRET efficiency calculated? (Would the sensor detach before it stretches to its maximum?)

This is an important point. Unfortunately, and to our best knowledge the unbinding force has not been characterized for lamin nanobody. However, for anti-EGFP nanobody the measured unbinding force between the nanobody and EGFP was over 41 - 56 pN (<https://iopscience.iop.org/article/10.1088/1478-3975/12/5/056009>). The unbinding force is known to be highly dependent on the loading rate and most probably depends

on nanobody – epitope pair, but the measured value suggests that the TSm_{od} will unfold before the nanobody is dissociated due to the force. We have added a short discussion related to the topic (lines 110-113) in the article.

- Was there any special reason why the authors used MDCK cells? If yes, please add.

There was no specific reason to select the MDCK cells. Both labs have interest towards epithelial mechanics and continuously work with MDCK cells.

- According to the methods section the authors used Airyscan microscopy to obtain images with a higher resolution. Can you indicate on the figure caption which images were acquire with this microscope?

This is a good suggestion. We have added the information to the figure captions.

- Is the panel b in Figure 1 a xy-projection of a z-stack? If yes, can you add this information on the figure caption?

The data represents a single section from the middle of the nucleus to highlight the nuclear lamina localization of the lamin antibody and the sensors. We have added a more specific description to the figure legend.

- For the images evaluating overlap (Figs. 1b,3f, 3g, 4d), would it be possible to perform so image correlation in addition to line profiles? For figures 3f and 3g, it is not clear what is plotted in the graphs (c-term / rod-domain).

Thanks for the suggestion. We clarified the analysis and added scatter plots of the fluorescence data showing the correlation of the immunostaining and sensor intensities. The c-term. / rod-domain graph is the intensity ratio of these labels. We clarified that in the figure and in the text (lines 268-269).

- Figure 1e reports intensity but uses the same colormaps as the one used for FRET. For clarity, I would change this to grey or cyan colormap (as used for the other 'intensity-based' images).

Important suggestion which increases clarity, we changed the look-up-table to grayscale in the FRAP data in Figure 1 and Supplemental figure S11.

- Why is the distribution of the apparent FRET efficiency in supporting figure S1 broader than in Figure 1c? They are both for lamin-TS expressed in MDCK WT, no? Is this linked to the analysis method?

The most likely reason for the difference is the expression of the sensors. In old supplemental figure S1 (current supplemental figure S4) the sensor was transiently expressed, but in figure 1c we used cell lines with stable sensor expression. In addition, the analysis methodology was also different, in Figure 1c we used sFRET and in supplemental figure S4 riFRET.

- I miss information on how the nuclear volume was calculated – can you add this information on the methods section?

We have now added the description of the volume calculation to the manuscript (lines 700-704).

- For the experiment with the changes on the osmotic pressure – what is the effect of this treatment on the TM sensor?

This is an interesting question. However, we did not conduct these experiments, since we believe that the Lamin-TM sensor does not influence nuclear mechanics. In addition, due to the variation in nuclei sizes and morphologies would require extensive studies to distinguish differences between Lamin-SS and -TM expressing cells in this respect.

- Is the data shown in figure 1g coming from the same cell? If so, there is a small difference in the apparent FRET efficiency but a big difference in the morphology of the nucleus. I would expect that these morphological changes would be reflected in the FRET value measured – can the authors comment on this?

Reviewer probably means figure 2g. Yes, the data comes from the same cell. However, when we compared the nucleus shape to measured FRET values, we did not detect direct correlation. We hypothesize that the overall strain state of the lamina does not change substantially and thus the average FRET does not change. Unfortunately, currently we lack the methods to acquire high signal-to-noise FRET data, which would allow more accurate spatial mapping of the FRET values.

- On the beginning of the section regarding cell cycle, EMT and chromatin condensation, the authors wrote “Detecting a large heterogeneity in the quantified FRET values of Lamin-SS between individual cells (line 151)” – which data does this sentence refer to?

The sentence refers to the data in figures 1, 2 and 3. We have now clarified this sentence and added more specific reference to the data (lines 231-236).

- What was the rationale to evaluate the effect of EMT on nuclear lamin? This was not clear from the text (section starting on line 160).

Reviewer is correct that this was not clearly explained. EMT is known to affect cellular mechanics and induce actomyosin contractility along with other changes in e.g., in cell migration and cancer cell metastasis and invasion potential, and chromatin organization (<https://doi.org/10.1038%2Fnmr3758> and <https://doi.org/10.1038/nsmb.2084>). We applied TGF- β -mediated induction of EMT, which has previously shown to be a strong driver of EMT in epithelial cells (<https://doi.org/10.1038/cr.2009.5>). We now included text regarding the motivation of the study (lines 244-248).

- On line 179 the authors refer to “ratiometric imaging” to quantify the antibody labelling ratio. It was not clear how this analysis was done. In addition, I would try to analyze the images using image correlation methods (eg. Pearson correlation) – see previous comment.

The reviewer is correct that the ratiometric imaging needs to be clarified. The aim of the ratiometric imaging was to probe the lamina structure, since the antibody epitope exposure and thus the staining efficiency depends on the organization of lamins (<https://doi.org/10.1038/nmat4389> and <https://doi.org/10.7554/eLife.58541>). By comparing the intensity of the lamina organization sensitive antibody staining to more general lamin A/C antibody, we can follow the general accessibility of the nuclear lamina. We have added more details about the analysis to the results section (lines 274-275). Also, the explanation for ratiometric analysis was included into the materials & methods section (Lines 579-581).

- Line 202: “This sensor [Lamin-histone-SS] exhibited a more pronounced predominant nucleoplasmatic localization” – what do the authors mean with this?

We tried to emphasize the distribution difference of Lamin-SS and Lamin-histone-SS. We have clarified the text in that paragraph (lines 298-301).

- Line 205: “FRET efficiency of 17% (mean 0.27...)” is this correct? Looking at the figure (4e) and mean value, I would say it is 27%...

We thank the reviewer for careful reading of the manuscript and pointing this out this error. We corrected the value (27%).

- Line 226: I would not say that the authors “demonstrate that the lamin A/C network experiences significant force, which is regulated by actin, myosin, and the nuclear-cytoskeletal connections”. I think the authors show that is ‘force’ is affected by these parameters, but not that it is regulated (small detail). [‘force’ or distance...]

The reviewer is correct, we have changed the word regulated to affected (line 334).

- Line 248: “our histone-lamin sensor shows that nucleoplasmic lamins, as opposed to lamins at the periphery, are the primary lamin component mechanically interacting with chromatin”. I think the interaction with the chromatin is caused by the presence of the nanobody against histone in the sensor (the sensor will only be present where there is chromatin), and therefore the authors cannot say that the lamin at the periphery does not interact with chromatin (there might be a physical interaction).

Reviewer is correct. We have modified this part in the manuscript (lines 359-364).

- Line 287: “it is possible that mechanical forces applied across the sensor causes one or more of the fluorescent proteins to unfold”. I find this unlikely, especially given the reversible nature of the binding of the nanobody. I would expect that the sensor would unbind rather than unfold (it would be important to have the values of the forces required for unbinding and unfolding to support this hypothesis). Also – what would unfold first: the nanobody or the fluorescent protein?

Reviewer is correct that sensor unbinding force is probably lower than the force needed for complete unfolding of fluorescent protein (FP). Force for EYFP unfolding has been reported to be close to 70 pN (<https://doi.org/10.1074/jbc.M609890200>), which is higher than the reported unbinding force between anti-EGFP nanobody and EGFP (41 - 56 pN, see above). However, even smaller forces can lead into changes in optical properties of the FPs, including loss of fluorescence (<https://doi.org/10.1073/pnas.1704937114>). In addition, the unfolding and unbinding force depends on the proteins in question and on the force loading rate, thus we can't totally exclude the possibility of at least partial fluorescent protein unfolding. We have strengthened and clarified the argument in the manuscript. Unfortunately, we did not find studies regarding force sensitivity of the nanobodies and therefore the direct comparison of nanobody vs. FP force sensitivity is difficult.

- On the same sentence, the authors say that unfolding of the proteins would result in decreased fluorescence of the protein (agreed) and therefore reduced FRET - not sure if I agree 100% with this. I think this depends on how the FRET is calculated (lifetime-based FRET analysis would not be affected) and which of the proteins, donor or acceptor, unfolds.

We thank the reviewer for highlighting the protein unfolding and FRET issue. This is an important aspect of force sensors. We have used 3 different methods to quantify the

FRET and depending on the measurement strategy, the protein unfolding (donor or acceptor) would have different effects on the FRET efficiency.

In the case of FLIM measurements, unfolding of the donor would prevent the actual measurement of donor lifetime, if we assume that the process leads into total loss of functionality of the donor. Unfolding of the acceptor would lead to reduced FRET (if the acceptor functionality is lost). However, in this case the reduced FRET would reflect the high force experienced by the sensor and the effect (reduced FRET) would be similar as in the case when the sensor is extended.

In the case of intensity dependent measurements, the donor or acceptor unfolding could have different outcomes, depending on which of the protein is losing its fluorescence. Donor unfolding would be easily interpreted as increased FRET (low donor signal vs. high acceptor signal) but the acceptor unfolding would have the opposite effect (high donor signal vs. low acceptor signal). Thus, we agree with the reviewer that the fluorescent protein unfolding could influence the FRET in different ways. We have modified the text in the manuscript (lines 414-421).

- At the end of the discussion, the authors say that these strategies could be use with nanobody targeted to GFP and RFP proteins. But, in that case, what would be the advantage compared to intramolecular sensors? I thought that the biggest advantage of the sensors developed here was to avoid adding fluorescent proteins to specific proteins, as this could affect their functionality (line 58/59)

Yes, reviewer is correct that the main benefit of the sensor would be lost. However, the size of TSmod is double to EGFP and for force measurements TSmod needs to be fused internally to the protein in question. In many cases this is not feasible. In addition, currently we have only a handful of nanobodies which can be used in living cells. However, the large family of functional EGFP tagged proteins have already been used for live cell microscopy and other experiments. Thus, a generalized nanobody-based sensor against EGFP (or other tags) could be used with many of the previously established EGFP tagged proteins, allowing the force measurements between these proteins. We have improved the discussion related to the possibilities of the sensor (lines 430-436).

- For the data analysis after the treatments (with CytoD, Y-27632, sucrose...), what was the time between addition of treatment and data acquisition? 20 minutes in all the cases?

We thank the reviewer for noticing this lack of information in the manuscript. The sucrose treatment was conducted for 15 minutes prior to the imaging. Cytochalasin D treatment (current supplemental figure S7) was conducted "live" and cytochalasin D was added at 3 min time point. The graph (supplemental figure S7) is from the final 45 min time point. ROCK inhibition in Figure 3a was conducted for 60 min and in Figure 3c it was conducted "live" and time corresponds to the ROCK inhibition time. We have now clarified these aspects in the manuscript.

- To evaluate if chromatin condensation affect the lamin A/C organization, the authors used two different antibodies that recognized two different epitopes. Strangely, on figure 4f, there seems to be a link between the distribution of the antibody against the rod domain and the expression of the lamin-SS sensor: there are 3 cells in the ROI that are not expressing the SS sensor; in this cells, the anti-rod antibody seems to be more localized on the periphery of the nucleus. Is this a coincidence? Could the authors show/analyse the distribution of the anti-rod AB (in comparison with the C-term) in WT cells and cells expressing the lamin-SS sensor? This might give an indication of where the nanobody binds...

We thank the reviewer for this suggestion. We carefully analyzed the correlation between the intensity of different antibodies and the Lamin-SS sensor and found indeed a negative correlation between rod-domain antibody (EP4520) and Lamin-SS. Thus, as the expression of Lamin-SS increases, the labeling of the nuclear lamina by the antibody decreases. This was not observed with 2 other lamin A/C antibodies or lamin B1 antibody, used as a control. By conducting additional FRET experiments with lamin A/C truncates we were able to show that lamin nanobody binds to the region near the rod-domain antibody epitope. We have added a supplementary figure about this and discuss the results in the manuscript (lines 99-109, Supplemental figure S2).

- On figure 4g, there seems to be a statistically significant difference between the SS sensor measured in KO cells and the TM sensor (in both WT and KO cells). Could the authors explain where this difference might come from?

Reviewer is correct and this was observed also by the reviewer #2. We think that the difference rises from the constructs themselves, different cell lines, and sensor transfections. Lleres et al. were able to use FRET between histone fluorescent proteins to measure the compaction state of the chromatin (<https://doi.org/10.1083%2Fjcb.200907029>). Thus, the monovalent Histone-LaminA sensor might lead into higher intermolecular FRET between only histone binding sensors. In addition, the two nanobodies in the full-length version of the sensor might also limit the free movement of the fluorescent proteins and in monovalent sensor the proteins (donor and acceptor) are more mobile, leading to higher FRET. We have added a short discussion of this in the end of the results section (lines 310-314).

On a small note, I would like to thank the authors for immediately sharing these sensors with the scientific community by depositing them in Addgene.

We thank the reviewer for this comment, this was our goal from the beginning. Reviewer #1 also highlighted the importance of sharing the sensors with the scientific community. Unfortunately, our actual plan to distribute the sensors via Addgene had to be changed. Company Proteintech (previously Chromotek) contacted us regarding the sensor (after publication of bioRxiv preprint) and stated that they own the intellectual property rights of the used nanobodies and argued that based on the end user license agreement (EULA) we can't distribute the sensor freely. However, Proteintech understood well our needs and we have now reached a tentative agreement where Proteintech will distribute the

sensors with non-profit basis (charging a small fee to recover distribution costs). This ensures that the sensors will be easily accessible to the scientific community and allowing Proteintech to preserve their intellectual property rights. Thus, similarly to the paper Virant et al. (<https://doi.org/10.1038/s41467-018-03191-2>) we added the following text to the manuscript (lines 450-452):

“The nanobody was previously developed by Rothbauer et al²⁴ and the expression vector coding for the lamin-nanobody was provided by ChromoTek & Proteintech (Planegg-Martinsried, Germany) with a corresponding end user license agreement.”

REVIEWERS' COMMENTS

Reviewer #4 (Remarks to the Author):

The authors have addressed all of my concerns in the revised version of their manuscript. I'm aware that my comments weren't the easiest to address, but I applaud their efforts in addressing them all in a constructive manner.

I believe the article has been substantially improved and now meets the criteria for publication. I'm confident that many researchers will benefit from the new type of sensors presented in this work.

Furthermore, I'm pleased to hear that the review report will also be published. There were some interesting discussions there that, while somewhat outside the scope of the work, could be valuable for others working on FRET-based (force) sensors or nanobodies.